# Probably Approximately Correct Constrained Learning

**Luiz F. O. Chamon**
Dept. of Electrical and Systems Engineering
University of Pennsylvania
Pennsylvania, USA
`luizf@seas.upenn.edu`

**Alejandro Ribeiro**
Dept. of Electrical and Systems Engineering
University of Pennsylvania
Pennsylvania, USA
`aribeiro@seas.upenn.edu`

## Abstract

As learning solutions reach critical applications in social, industrial, and medical domains, the need to curtail their behavior has become paramount. There is now ample evidence that without explicit tailoring, learning can lead to biased, unsafe, and prejudiced solutions. To tackle these problems, we develop a generalization theory of constrained learning based on the probably approximately correct (PAC) learning framework. In particular, we show that imposing requirements does not make a learning problem harder in the sense that any PAC learnable class is also PAC *constrained* learnable using a constrained counterpart of the empirical risk minimization (ERM) rule. For typical parametrized models, however, this learner involves solving a constrained non-convex optimization program for which even obtaining a feasible solution is challenging. To overcome this issue, we prove that under mild conditions the empirical dual problem of constrained learning is also a PAC constrained learner that now leads to a practical constrained learning algorithm based solely on solving unconstrained problems. We analyze the generalization properties of this solution and use it to illustrate how constrained learning can address problems in fair and robust classification.

## 1 Introduction

Learning has become a core component of the modern information systems we increasingly rely upon to select job candidates, analyze medical data, and control "smart" applications (home, grid, city). As these systems become ubiquitous, so does the need to curtail their behavior. Left untethered, they can fail catastrophically as evidenced by the growing number of reports involving biased, prejudiced models or systems prone to tampering (e.g., adversarial examples), unsafe behaviors, and deadly accidents [1–6]. Typically, learning is constrained by using domain expert knowledge to either construct models that *embed* the required properties (see, e.g., [7–13]) or *tune* the training objective so as to promote them (see, e.g., [14–17]). The latter approach, known as regularization, is ubiquitous in practice even though it need not yield feasible solutions [18]. In fact, existing results from classical learning theory guarantee generalization with respect to the regularized objective, which says nothing about meeting the requirements it may describe [19, 20]. While the former approach guarantees that the solution satisfies the requirements, the scale and opacity of modern machine learning (ML) systems render this model design impractical.

Since ML models are often trained using empirical risk minimization (ERM), an alternative solution is to explicitly add constraints to these optimization problems. Since requirements are often expressed as constraints in the first place, this approach overcomes the need to tune regularization parameters. What it more, any solution automatically satisfies the requirements. Nevertheless, this approach suffers from two fundamental drawbacks. First, its involves solving a constrained optimization problem that is non-convex for typical parametrizations (e.g., neural networks). Though gradient descent can often be used to obtain good minimizers for differentiable models, it does not guarantee

constraint satisfaction. Indeed, there is typically no straightforward way to project onto the feasibility set (e.g., the set of fair classifiers) and strong duality need not hold for non-convex programs [18]. Second, even if we could solve this constrained ERM, the issue remains of how its solutions generalize since classical learning theory is involved only with unconstrained problems [19, 20].

In this work, we address these issues in two steps. We begin by formalizing the concept of constrained learning using the probably approximately correct (PAC) framework. We prove that any hypothesis class that is unconstrained learnable is constrained learnable and that the constrained counterpart of the ERM rule is a PAC constrained learner. Hence, we establish that, from a learning theoretic perspective, *constrained learning is as hard as unconstrained (classical) learning*. This, however, does not resolve the practical issue of learning under requirements due to the non-convexity of the constrained ERM problem. To do so, we proceed by deriving an empirical saddle-point problem that is a (representation-independent) PAC constrained learner. We show that its approximation error depends on the richness of the parametrization and the difficulty of satisfying the learning constraints. Finally, we put forward practical constrained learning algorithm that we use to illustrate how constrained learning can address problems involving fairness and robustness.

## 2   Related work

Central to ML is the concept of ERM in which statistical quantities are replaced by their empirical counterparts, thus allowing learning problems to be solved from data, without prior knowledge of its underlying distributions. The set of conditions under which this is a sensible approach is known in learning theory as (agnostic) PAC learnability. More generally, the PAC framework formalizes what it means to solve a statistical learning problem and studies when it can be done [19–22]. While different learning models, such as structured complexity and PAC-Bayes, have been proposed, they are beyond the scope of this work.

The objects studied in (PAC) learning theory, however, are unconstrained statistical learning problem. Yet, there is a growing need to enable learning under constraints to tackle problems in fairness [23–29], robustness [30–32], safety [33–37], and semi-supervised learning [38–40], to name a few. While constraints have been used in statistics since Neyman-Pearson [41], generalization guarantees for constrained learning have been studied only in specific contexts, e.g., for coherence constraints or rate-constrained learning [23, 25, 29, 42]. Additionally, due to the non-convexity of typical learning problems, many of these results hold for randomized solutions, e.g., [23, 25, 27, 29]. In contrast, this work puts forward a formal constrained learning framework in which generalization results are derived for deterministic learners. A first step in that direction was taken in [43], albeit from an optimization perspective. This work also accounts for pointwise constraints, fundamental in the context of fairness, and provides a practical, guaranteed constrained learning algorithm (Sec. 5.2).

Due to these challenges, learning under requirements is often tackled using regularization, i.e., by integrating a fixed cost for violating the constraints into the training objective (see, e.g., [15–17, 31, 44, 45]). Selecting these costs, however, can be challenging, especially as the number of constraints grows. In fact, their values often depend on the problem instance, the objective value, and can interact in non-trivial ways [46–50]. In the case of convex optimization problems, a straightforward relation between constraints and regularization costs can be obtained due to strong duality. A myriad of primal-dual methods can then be used to obtain optimal, feasible solutions [51]. However, most modern parametrizations (e.g., CNNs) lead to non-convex programs for which a regularized formulation need not yield feasible solutions, all the more so good ones [18]. While primal-dual algorithms have been used in practice, no guarantees can be given for their outcome in general [30, 32, 52, 53].

## 3   Constrained Learning

Let $\mathfrak{D}_i$, $i = 0, \dots, m+q$, denote *unknown* probability distributions over the space of data pairs $(\boldsymbol{x}, y)$, with $\boldsymbol{x} \in \mathcal{X} \subset \mathbb{R}^d$ and $y \in \mathcal{Y} \subset \mathbb{R}$. For a hypothesis class $\mathcal{H}$ of functions $\phi : \mathcal{X} \to \mathbb{R}^k$, define the

generic constrained statistical learning (CSL) problem as

$$P^\star = \min_{\phi \in \mathcal{H}} \quad \mathbb{E}_{(\boldsymbol{x},y)\sim\mathfrak{D}_0}\Big[\ell_0\big(\phi(\boldsymbol{x}),y\big)\Big]$$

$$\text{subject to} \quad \mathbb{E}_{(\boldsymbol{x},y)\sim\mathfrak{D}_i}\Big[\ell_i\big(\phi(\boldsymbol{x}),y\big)\Big] \le c_i, \quad i = 1, \ldots, m, \qquad\qquad \text{(P-CSL)}$$

$$\ell_j\big(\phi(\boldsymbol{x}),y\big) \le c_j \quad \mathfrak{D}_j\text{-a.e.}, \qquad j = m+1, \ldots, m+q,$$

where $\ell_i : \mathbb{R}^k \times \mathcal{Y} \to \mathbb{R}$ are performance metrics. In general, we think of $\mathfrak{D}_0$ as a nominal joint distribution over data pairs $(\boldsymbol{x},y)$ corresponding to feature vectors $\boldsymbol{x}$ and responses $y$. The additional $\mathfrak{D}_i$ can be used to model different conditional distributions over which requirements are imposed either on average, through the losses $\ell_i$, $i \le m$, or pointwise, through the losses $\ell_j$, $j > m$. Note that the unconstrained version of (P-CSL), namely

$$P_U^\star = \min_{\phi \in \mathcal{H}} \quad \mathbb{E}_{(\boldsymbol{x},y)\sim\mathfrak{D}_0}\Big[\ell_0\big(\phi(\boldsymbol{x}),y\big)\Big], \qquad\qquad \text{(PI)}$$

is at the core of virtually all of modern ML [20, 54].

Before tackling *if* and *how* we can learn under constraints, i.e., whether we can solve (P-CSL), we illustrate *what* constrained learning can enable. To make the discussion concrete, we present two constrained formulations of the learning problems we solve in Section 6.

**Invariance and fair learning.** Constrained learning is a natural way to formulate learning problems in which invariance is required. Consider a model $\phi$ whose output is a discrete distribution over $k$ possible classes. Then, (P-CSL) can be used to write

$$\underset{\phi \in \mathcal{H}}{\text{minimize}} \quad \mathbb{E}_{(\boldsymbol{x},y)\sim\mathfrak{D}}\Big[\ell_0\big(\phi(\boldsymbol{x}),y\big)\Big]$$

$$\text{subject to} \quad \mathbb{E}_{(\boldsymbol{x},y)\sim\mathfrak{D}}\Big[\mathrm{D}_{\mathrm{KL}}\big(\phi(\boldsymbol{x}) \,\|\, \phi(\rho(\boldsymbol{x}))\big)\Big] \le c, \qquad\qquad \text{(PII)}$$

where $\rho$ is an input transformation we wish the model to be invariant to and $c > 0$ determines the sensitivity level. Formulation (PII) can be extended trivially to multiple transformations (see Sec. 6). When the average invariance in (PII) is not enough, a stricter, pointwise requirement can be imposed, by using

$$\mathrm{D}_{\mathrm{KL}}\big(\phi(\boldsymbol{x}) \,\|\, \phi(\rho(\boldsymbol{x}))\big) \le c \quad \mathfrak{D}\text{-a.e.}. \qquad\qquad (1)$$

For instance, fairness can be seen as a form of invariance in which $\rho$ induces an alternative distribution of a certain protected variable (e.g., a gender change) [23–26, 28, 29]. In this case, the constraint in (PII) is related to the average causal effect (ACE) and (1) to *counterfactual fairness* [24]. While fairness goes beyond invariance, our goal is not to litigate the merit of any fairness metrics, but to show how constrained learning may provide a natural way to encode them.

**Robust learning.** Another issue affecting ML models, especially CNNs, is robustness. It is straightforward to construct small input perturbations that lead to misclassification and there are now numerous methods to do so. While adversarial training has empirically been shown to improve robustness, it often results in classifiers with poor nominal performance [30, 31, 44, 52, 53, 55]. In [32], a constrained formulation involving an upper bound on the worst-case error was used to tackle this issue. Similarly, we can address this compromise using (P-CSL) by writing

$$\underset{\phi \in \mathcal{H}}{\text{minimize}} \quad \mathbb{E}_{(\boldsymbol{x},y)\sim\mathfrak{D}}\Big[\ell_0\big(\phi(\boldsymbol{x}),y\big)\Big]$$

$$\text{subject to} \quad \mathbb{E}_{(\boldsymbol{x},y)\sim\mathfrak{A}}\Big[\ell_0\big(\phi(\boldsymbol{x}),y\big)\Big] \le c \qquad\qquad \text{(PIII)}$$

where $\mathfrak{A}$ is an adversarial data distributions. What is more, we can soften the worst-case requirements of robust optimization by taking $\mathfrak{A} \mid \varepsilon$ to be a distribution of adversarials with perturbation at most $\varepsilon$ and pose a prior on $\varepsilon$ (e.g., an exponential). This results in classifiers whose performance degrades smoothly with the perturbation magnitude. The theory and algorithms developed in this work give generalization guarantees on solutions of this problem obtained using samples of $\mathfrak{A}$, which can be accessed based on, e.g., adversarial attacks (Sec. 6). In other words, it establishes conditions under which a classifier that is accurate and robust during training is also accurate and robust during testing.

# 4 Probably Approximately Correct Constrained Learning

While (P-CSL) clearly addresses many of the issues discussed in Sec. 1, we cannot expect to solve it exactly without access to the $\mathfrak{D}_i$ against which expectations are evaluated. Additionally, solving the variational (P-CSL) is challenging unless $\mathcal{H}$ is finite. In this section, we address the first matter by settling, as in classical learning theory, on obtaining a *good enough* solution (Sec. 4.1). We then show that these solutions are not "harder" to get in constrained learning than they were in unconstrained learning (Sec. 4.2). We then proceed to tackle the algorithmic challenges by deriving and analyzing a practical constrained learning algorithm (Sec. 5.2).

## 4.1 From PAC to PACC

Let us begin by defining what it means to learn under constraints. To do so, we start by looking at the unconstrained case, which is addressed in learning theory under the PAC framework [19–22].

**Definition 1** (PAC learnability). *A hypothesis class $\mathcal{H}$ is* (agnostic) probably approximately correct (PAC) *learnable if for every $\epsilon, \delta \in (0, 1)$ and every distribution $\mathfrak{D}_0$, a $\phi^\dagger \in \mathcal{H}$ can be obtained from $N \geq N_{\mathcal{H}}(\epsilon, \delta)$ samples of $\mathfrak{D}_0$ such that $\mathbb{E}\left[\ell_0\big(\phi^\dagger(\boldsymbol{x}), y\big)\right] \leq P_U^\star + \epsilon$ with probability $1 - \delta$.*

A classical result states that $\mathcal{H}$ is PAC learnable if and only if it has finite VC dimension and that the $\phi^\dagger$ from Def. 1 can be obtained by solving an ERM problem [19, 20]. This is, however, not enough to enable constrained learning since a PAC $\phi^\dagger$ may not be feasible for (P-CSL). In fact, feasibility often takes priority over performance in constrained learning problems. For instance, regardless of how good a fair classifier is, it serves no "fair" purpose in practice unless it meets fairness requirements [see, e.g., (PII)]. These observations lead us to the following definition.

**Definition 2** (PACC learnability). *A hypothesis class $\mathcal{H}$ is* probably approximately correct constrained (PACC) *learnable if for every $\epsilon, \delta \in (0, 1)$ and every distribution $\mathfrak{D}_i$, $i = 0, \dots, m + q$, a $\phi^\dagger \in \mathcal{H}$ can be obtained based $N \geq N_{\mathcal{H}}(\epsilon, \delta)$ samples from each $\mathfrak{D}_i$ such that it is, with probability $1 - \delta$,*

*1) approximately optimal, i.e.,*

$$\mathbb{E}_{(\boldsymbol{x},y)\sim\mathfrak{D}_0}\left[\ell_0\big(\phi^\dagger(\boldsymbol{x}), y\big)\right] \leq P^\star + \epsilon \quad \text{and} \tag{2}$$

*2) approximately feasible, i.e.,*

$$\mathbb{E}_{(\boldsymbol{x},y)\sim\mathfrak{D}_i}\left[\ell_i\big(\phi^\dagger(\boldsymbol{x}), y\big)\right] \leq b_i + \epsilon, \qquad\qquad i = 1, \dots, m, \tag{3a}$$

$$\ell_j\big(\phi^\dagger(\boldsymbol{x}), y\big) \leq b_j, \; \text{for all } (\boldsymbol{x}, y) \in \mathcal{K}_j, \quad j = m+1, \dots, m+q, \tag{3b}$$

*where $\mathcal{K}_j \subseteq \mathcal{X} \times \mathcal{Y}$ are sets of $\mathfrak{D}_j$ measure at least $1 - \epsilon$.*

Note that every PACC learnable class is also PAC learnable since it satisfies (2). However, a PACC learner must also meet the probably approximate feasibility conditions in (3). The additional "C" in PACC is used to remind ourselves of this fact. Next, we show that the converse is also true, i.e., that PAC and PACC learning are equivalent problems.

## 4.2 PACC Learning is as Hard as PAC Learning

Having formalized what we mean by constrained learning (Sec. 4.1), we turn to the issue of when it can be done. To do so, we follow the unconstrained learning lead and put forward an empirical constrained risk minimization (ECRM) rule using $N_i$ samples $(\boldsymbol{x}_{n_i}, y_{n_i}) \sim \mathfrak{D}_i$, namely

$$\hat{P}^\star = \min_{\phi \in \mathcal{H}} \quad \frac{1}{N_0} \sum_{n_0=1}^{N_0} \ell_0\big(\phi(\boldsymbol{x}_{n_0}), y_{n_0}\big)$$

$$\text{subject to} \quad \frac{1}{N_i} \sum_{n_i=1}^{N_i} \ell_i\big(\phi(\boldsymbol{x}_{n_i}), y_{n_i}\big) \leq c_i, \qquad i = 1, \dots, m \tag{P-ECRM}$$

$$\ell_j\big(\phi(\boldsymbol{x}_{n_j}), y_{n_j}\big) \leq c_j, \text{for all } n_j, \quad j = m+1, \dots, m+q.$$

Notice that (P-ECRM) is a constrained version of the classical ERM problem that is ubiquitous in the solution of unconstrained learning problems [20, 54]. The next theorem shows that, under mild assumptions on the losses, if $\mathcal{H}$ is PAC learnable, then it is PACC learnable using (P-ECRM).

**Theorem 1.** *Let the $\ell_i$, $i = 0, \ldots, m + q$, be bounded on $\mathcal{X}$. The hypothesis class $\mathcal{H}$ is PACC learnable if and only if it is PAC learnable and (P-ECRM) is a PACC learner of $\mathcal{H}$. Explicitly, let $d_{\mathcal{H}} < \infty$ be the VC dimension of $\mathcal{H}$. If $N_i \geq C\zeta^{-1}(\epsilon, \delta, d_{\mathcal{H}})$, $i = 0, \ldots, m + q$, for an absolute constant $C$ and*

$$\zeta^{-1}(\epsilon, \delta, d) = \frac{d + \log(1/\delta)}{\epsilon^2}, \tag{4}$$

*then any solution $\hat{\phi}^\star$ of (P-ECRM) is a PACC solution of (P-CSL).*

*Proof.* See Appendix A in the extended version [56]. $\square$

Theorem 1 shows that, from a learning theoretic point-of-view, constrained learning is as hard as unconstrained learning. Not only that, but notice the sample complexity of constrained described by (4) matches that of PAC learning [19, 20]. It is therefore not surprising that a constrained version of ERM is a PACC learner. A similar result appeared in [26] for a particular rate constraint and not in the context of PACC learning. Still, solving (P-ECRM) remains challenging. Indeed, while it addresses the statistical issue of (P-CSL), it remains, in most practical cases, an infinite dimensional (functional) problem. This issue is often addressed by leveraging a finite dimensional parametrization of (a subset of) $\mathcal{H}$, such as a kernel model or a (C)NN. Explicitly, we associate to each parameter vector $\boldsymbol{\theta} \in \mathbb{R}^p$ a function $f_{\boldsymbol{\theta}} \in \mathcal{H}$, replacing (P-ECRM) by

$$
\begin{aligned}
\hat{P}_{\theta}^\star = \min_{\boldsymbol{\theta} \in \mathbb{R}^p} \quad & \frac{1}{N_0} \sum_{n_0=1}^{N_0} \ell_0\big(f_{\boldsymbol{\theta}}(\boldsymbol{x}_{n_0}), y_{n_0}\big) \\
\text{subject to} \quad & \frac{1}{N_i} \sum_{n_i=1}^{N_i} \ell_i\big(f_{\boldsymbol{\theta}}(\boldsymbol{x}_{n_i}), y_{n_i}\big) \leq c_i, \qquad i = 1, \ldots, m \\
& \ell_j\big(f_{\boldsymbol{\theta}}(\boldsymbol{x}_{n_j}), y_{n_j}\big) \leq c_j, \text{ for all } n_j, \quad j = m + 1, \ldots, m + q.
\end{aligned}
\tag{PIV}
$$

Even if (P-ECRM) is a convex program in $\phi$, (PIV) typically is not a convex program in $\boldsymbol{\theta}$ (except, e.g., if the losses are convex and $f_{\boldsymbol{\theta}}$ is linear in $\boldsymbol{\theta}$). This issue also arises in unconstrained learning problems, but is exacerbated by the presence of constraints. Though it is sometimes possible to find good approximate minimizers of $\ell_0$ using, e.g., gradient descent rules [57–61], even obtaining a feasible $\boldsymbol{\theta}$ may be challenging. Indeed, although good CNN classifiers can be trained using gradient descent, obtaining a good *fair/robust* classifier is considerably harder. Regularized formulations are often used to sidestep this issue by incorporating a linear combination of the constraints into the objective and solving the resulting unconstrained problem [15–17, 31, 44, 45]. Nevertheless, whereas the generalization guarantees of classical learning theory apply to this modified objective, they say nothing of the requirements it describes. Since strong duality need not hold for the non-convex (PIV), this procedure need not be PACC (Def. 2) and may lead to solutions that are either infeasible or whose performance is unacceptably poor [18].

While no formal connection can be drawn between (PIV) and its regularized formulation (due to the lack of strong duality [18]), its dual problem turns out to be related to (P-CSL). In the sequel, we prove that it provides (near-)PACC solutions for (P-CSL) with an approximation error in (2) that depends on the richness of the parametrization and how strict the learning constraints are (Sec. 5.1). In fact, we show that it is a (near-)PACC learner even if the parametrization is PAC learnable but $\mathcal{H}$ is not. Based on this result, we obtain a practical constrained learning algorithm (Sec. 5.2) that we use to solve the problems formulated in Sec. 3.

## 5 A (Near-)PACC Learning Algorithm

In this section, we derive a practical constrained learning algorithm by first analyzing the dual problem of (PIV) (Sec. 5.1) and then proposing an algorithm to solve it (Sec. 5.2). Although we know this dual problem is not related to (PIV), we prove that it is related directly to the original constrained learning problem (P-CSL) by showing it is a PACC learner except for an approximation error determined by the quality of the parametrization. We formalize this concept as follows:

**Definition 3** (Near-PACC learnability)**.** *A class $\mathcal{H}$ is* (near-)PACC *learnable through a class $\mathcal{P}$ if there exists an $\epsilon_0 > 0$ such that for every $\epsilon, \delta \in (0, 1)$ and every distribution $\mathfrak{D}_i$, $i = 0, \ldots, m + q$,*

*an approximately feasible $\phi^\dagger \in \mathcal{P}$ [viz. (3)] can be obtained with probability $1 - \delta$ based on $N \geq N_{\mathcal{P}}(\epsilon, \delta)$ samples from each $\mathfrak{D}_i$ and $\mathbb{E}_{(\boldsymbol{x},y)\sim\mathfrak{D}_0}\left[\ell_0\left(\phi^\dagger(\boldsymbol{x}), y\right)\right] \leq P^\star + \epsilon_0 + \epsilon$*

In Def. 3, $\epsilon_0$ characterizes the *approximation error*. In contrast to unconstrained learning, however, this error cannot be separated from the learning problem due to the constraints. Still, it is *fixed*, i.e., it is independent of the sample set, and affects neither the sample complexity nor the constraint satisfaction. Hence, the parametrized constrained learner sacrifices optimality, but not feasibility, which remains dependent only on the number of samples $N$ (Def. 2). Finally, observe that the sample complexity does not depend on the original hypothesis class $\mathcal{H}$, but on the parametrized $\mathcal{P}$. Near-PACC is therefore related to representation-independent learning [20].

## 5.1 The Empirical Dual Problem of (P-CSL)

We begin by analyzing the gap between (P-CSL) and its (parametrized) empirical dual problem. Define the (parametrized) empirical Lagrangian of (P-CSL) as

$$
\hat{L}(\boldsymbol{\theta}, \boldsymbol{\mu}, \boldsymbol{\lambda}_j) = \frac{1}{N_0} \sum_{n_0=1}^{N_0} \ell_0\left(f_{\boldsymbol{\theta}}(\boldsymbol{x}_{n_0}), y_{n_0}\right) + \sum_{i=1}^{m} \mu_i \left[ \frac{1}{N_i} \sum_{n_i=1}^{N_i} \ell_i\left(f_{\boldsymbol{\theta}}(\boldsymbol{x}_{n_i}), y_{n_i}\right) - c_i \right]
$$
$$
+ \sum_{j=m+1}^{m+q} \left[ \frac{1}{N_j} \sum_{n_j=1}^{N_j} \lambda_{j,n_j} \left( \ell_j\left(f_{\boldsymbol{\theta}}(\boldsymbol{x}_{n_j}), y_{n_j}\right) - c_j \right) \right],
\tag{5}
$$

where $\boldsymbol{\mu} \in \mathbb{R}_+^m$ collects the dual variables $\mu_i$ relative to the average constraints and $\boldsymbol{\lambda}_j \in \mathbb{R}_+^{N_j}$ collects the dual variables $\lambda_{j,n_j}$ relative to the $j$-th pointwise constraint. The empirical dual problem of (P-CSL) is then written as

$$
\hat{D}^\star = \max_{\boldsymbol{\mu}\in\mathbb{R}_+^m, \, \boldsymbol{\lambda}_j\in\mathbb{R}_+^{N_j}} \min_{\boldsymbol{\theta}\in\mathbb{R}^p} \hat{L}(\boldsymbol{\theta}, \boldsymbol{\mu}, \boldsymbol{\lambda}_j),
\tag{$\hat{\mathrm{D}}$-CSL}
$$

Note that ($\hat{\mathrm{D}}$-CSL) is the dual problem of the parametrized ECRM (PIV). However, due to its non-convexity, its holds only that $\hat{D}^\star \leq \hat{P}_{\boldsymbol{\theta}}^\star$ and, in general, a saddle-point of ($\hat{\mathrm{D}}$-CSL) is not related to a solution of (PIV) [18]. Still, ($\hat{\mathrm{D}}$-CSL) can be related directly to (P-CSL), which is why we refer to it as its empirical dual. This relation obtains under the following assumptions:

**Assumption 1.** The losses $\ell_i(\cdot, y)$, $i = 0, \ldots, m + q$, are $[0, B]$-valued, $M$-Lipschitz, convex functions for all $y \in \mathcal{Y}$. The loss $\ell_0$ is additionally strongly convex.

**Assumption 2.** The hypothesis class $\mathcal{H}$ is convex, the parametrized $\mathcal{P} = \{f_{\boldsymbol{\theta}} \mid \boldsymbol{\theta} \in \mathbb{R}^p\} \subseteq \mathcal{H}$ is PAC learnable, and there is $\nu > 0$ such that for each $\phi \in \mathcal{H}$ there exists $f_{\boldsymbol{\theta}} \in \mathcal{P}$ for which $\sup_{\boldsymbol{x}\in\mathcal{X}} |f_{\boldsymbol{\theta}}(\boldsymbol{x}) - \phi(\boldsymbol{x})| \leq \nu$.

**Assumption 3.** There exists $\boldsymbol{\theta}' \in \mathbb{R}^p$ such that $f_{\boldsymbol{\theta}'}$ is strictly feasible for (P-CSL) with constraints $c_i - M\nu$ and $c_j - M\nu$ and for each datasets $\mathcal{S} = \left\{(\boldsymbol{x}_{n_i}, y_{n_i})\right\}_{i=0,\ldots,m+q}$ there exists a $\boldsymbol{\theta}''$ that is strictly feasible for (PIV).

In contrast to the unconstrained learning setting or the ECRM result in Theorem 1, we require that the losses $\ell_i$ and the hypothesis class $\mathcal{H}$ be convex. This, however, does not imply that ($\hat{\mathrm{D}}$-CSL) or (PIV) are convex problems since $\ell_i(f_{\boldsymbol{\theta}}(\boldsymbol{x}), y)$ need not be convex in $\boldsymbol{\theta}$. Additionally, only the parametrized class $\mathcal{P}$ is required to be PAC learnable. Hence, $\mathcal{H}$ can be the space of continuous functions or a reproducing kernel Hilbert space (RKHS) and $f_{\boldsymbol{\theta}}$ can be a neural network [62–64] or a finite linear combinations of kernels [65, 66], both of which meet the uniform approximation assumption. This assumption can also be relaxed in the absence of pointwise constraints (Remark 1). Assumption 3 guarantees that the problem is well-posed, i.e., a feasible solution for (P-CSL) can be found in $\mathcal{P}$.

The main result of this section is collected in the following theorem.

**Theorem 2.** *Let $d_{\mathcal{P}}$ be the VC dimension of $\mathcal{P}$. Under Assumptions 1–3, ($\hat{\mathrm{D}}$-CSL) is a near-PACC learner of $\mathcal{H}$ with $N_{\mathcal{P}} = C\zeta^{-1}(\epsilon, \delta, d_{\mathcal{P}})$, for an absolute constant $C$ and $\zeta^{-1}$ as in (4), and*

$$
\epsilon_0 = \left(1 + \left\|\boldsymbol{\mu}_p^\star\right\|_1 + \left\|\boldsymbol{\lambda}_p^\star\right\|_{L_1}\right) M\nu,
\tag{6}
$$

*where $(\boldsymbol{\mu}_p^\star, \boldsymbol{\lambda}_p^\star)$ are dual variables of (P-CSL) with constraints $c_i - M\nu$ for $i = 1, \ldots, m + q$.*

---

**Algorithm 1** Primal-dual near-PACC learner

---

1: *Initialize*: $\boldsymbol{\theta}^{(0)} = 0$, $\boldsymbol{\mu}^{(0)} = \mathbb{1}$, $\boldsymbol{\lambda}_j^{(0)} = \mathbb{1}$
2: **for** $t = 1, \ldots, T$
3:     Obtain $\boldsymbol{\theta}^{(t-1)}$ such that $\hat{L}\left(\boldsymbol{\theta}^{(t-1)}, \boldsymbol{\mu}^{(t-1)}, \boldsymbol{\lambda}_j^{(t-1)}\right) \leq \min_{\boldsymbol{\theta} \in \mathbb{R}^p} \hat{L}\left(\boldsymbol{\theta}, \boldsymbol{\mu}^{(t-1)}, \boldsymbol{\lambda}_j^{(t-1)}\right) + \rho$
4:     Update dual variables

$$\mu_i^{(t)} = \left[\mu_i^{(t-1)} + \eta\left(\frac{1}{N_i}\sum_{n_i=1}^{N_i} \ell_i\big(f_{\boldsymbol{\theta}^{(t-1)}}(\boldsymbol{x}_{n_i}), y_{n_i}\big) - c_i\right)\right]_+$$

$$\lambda_{j,n_j}^{(t)} = \left[\lambda_{j,n_j}^{(t-1)} + \frac{\eta}{N_j}\Big(\ell_j\big(f_{\boldsymbol{\theta}^{(t-1)}}(\boldsymbol{x}_{n_j}), y_{n_j}\big) - c_j\Big)\right]_+$$

5: **end**

---

*Proof.* See Appendix B in the extended version [56].     □

Thus, the approximation error incurred by using the parametrization $f_{\boldsymbol{\theta}}$ is affected by (i) the difficulty of the learning problem and (ii) the richness of the parametrization. Indeed, under Assumptions 1–3, (P-CSL) is a strongly dual functional problem whose dual variables have a well-known sensitivity interpretation [67, Sec. 5.6]. So the bracketed quantity in (6) quantifies how stringent the learning constraints are in terms of how much performance could be gained by relaxing them. In addition, $\epsilon_0$ is affected by the approximation capability $\nu$ of the parametrization. Since better parametrizations typically involve more parameters, which in turn affects the VC dimension of $\mathcal{P}$, a typical compromise between the approximation error and complexity arises. For small sample sets, the generalization error in Def. 3 is dominated by the estimation error $\epsilon$, which improves for lower complexity classes. If there is abundance of data or the learning requirements are particularly stringent, the approximation error $\epsilon_0$ dominates and more accurate, even if more complex, parametrizations should be used.

Note that the dual variables $(\boldsymbol{\mu}_p^\star, \boldsymbol{\lambda}_p^\star)$ may be hard to evaluate since they are related to a version of the statistical problem (P-CSL). While their norms can be estimated using classical results from optimization theory (see, e.g., [18, 68]), they often lead to loose, uninformative bounds. Notice, however, that only $\epsilon$ depends on the sample size.

**Remark 1.** When the constrained learning problem has no pointwise constraints [$q = 0$ in (P-CSL)], Assumption 2 can be relaxed from a uniform to a total variation approximation. Explicitly, Theorem 1 holds if for each $\phi \in \mathcal{H}$ there exist $\boldsymbol{\theta} \in \mathbb{R}^p$ such that $\mathbb{E}_{(\boldsymbol{x},y) \sim \mathfrak{D}_i}\big[\|f_{\boldsymbol{\theta}}(\boldsymbol{x}) - \phi(\boldsymbol{x})\|\big] \leq \nu$ for all $i$.

### 5.2 A Primal-Dual near-PACC Learner

We now proceed to introduce a practical algorithm to solve (D̂-CSL) based on a (sub)gradient primal-dual method. To do so, start by noting that the outer maximization is a convex optimization program. Indeed, the dual function $\hat{d}(\boldsymbol{\mu}, \boldsymbol{\lambda}_j) = \min_{\boldsymbol{\theta}} \hat{L}(\boldsymbol{\theta}, \boldsymbol{\mu}, \boldsymbol{\lambda}_j)$ is the pointwise minimum of a set of affine functions and is therefore always concave [18]. Additionally, its (sub)gradients can be easily computed by evaluating the constraint slacks at the minimizer of $\hat{L}$ [51, Ch. 3]. Hence, the main challenge in (D̂-CSL) is the inner minimization.

Despite the Lagrangian (5) often being non-convex in $\boldsymbol{\theta}$, (D̂-CSL) is an unconstrained optimization problem. Hence, contrary to (PIV), it is often the case that good minimizers can be found, especially for differentiable losses and parametrizations (i.e., most common ML models). For instance, there is ample empirical and theoretical evidence that gradient descent can learn to good parameters for (C)NNs [57–61]. In that vein, we thus assume that we have access to the following oracle:

**Assumption 4.** There exists an oracle $\boldsymbol{\theta}^\dagger(\boldsymbol{\mu}, \boldsymbol{\lambda}_j)$ and $\rho > 0$ such that $\hat{L}\big(\boldsymbol{\theta}^\dagger(\boldsymbol{\mu}, \boldsymbol{\lambda}_j), \boldsymbol{\mu}, \boldsymbol{\lambda}_j\big) \leq \min_{\boldsymbol{\theta}} \hat{L}\big(\boldsymbol{\theta}, \boldsymbol{\mu}, \boldsymbol{\lambda}_j\big) + \rho$ for all $\boldsymbol{\mu} \in \mathbb{R}_+^m$ and $\boldsymbol{\lambda}_j \in \mathbb{R}_+^{N_j}$, $j = m+1, \ldots, m+q$.

Assumption 4 essentially states that we are able to (approximately) train regularized unconstrained learners using the parametrization $f_{\boldsymbol{\theta}}$. We can alternate between minimizing the Lagrangian (5) with

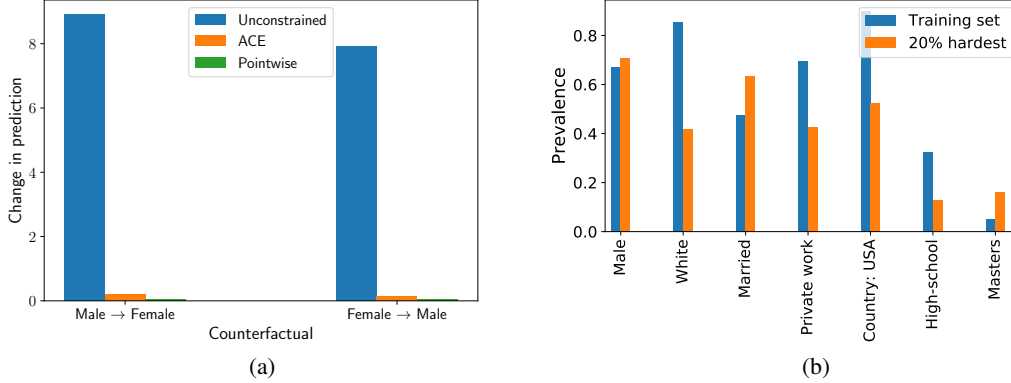

Figure 1: Fair classification (Adult dataset): (a) classifier sensitivity and (b) prevalence of different groups among the 20% training set examples with largest dual variables.

respect to $\boldsymbol{\theta}$ for fixed $(\boldsymbol{\mu}, \boldsymbol{\lambda}_j)$ and updating the dual variables using the resulting minimizer. This procedure is summarized in Algorithm 1 and analyzed in the following theorem:

**Theorem 3.** *Fix $\beta > 0$ and consider Algorithm 1 with at least $C\zeta^{-1}(\epsilon, \delta, d_{\mathcal{P}})$ samples from each $\mathfrak{D}_j$, where $C$ is an absolute constant, $\zeta^{-1}$ is as in (4), and $d_{\mathcal{P}}$ is the VC dimension of $\mathcal{P}$. Under Assumptions 1–4, Algorithm 1 converges to the neighborhood*

$$P^\star - \rho - \beta - \eta S - \epsilon \le \hat{L}\left(\boldsymbol{\theta}^{(T)}, \boldsymbol{\mu}^{(T)}, \boldsymbol{\lambda}_j^{(T)}\right) \le P^\star + \rho + \epsilon_0 + \epsilon \tag{7}$$

*with probability $1 - \delta$ after at most $T = O(1/\beta)$ for $\epsilon_0$ as in (6) and $S = O\big(B^2\big)$.*

*Proof.* See Appendix C in the extended version [56]. $\square$

Theorem 3 bounds the suboptimality of Algorithm 1 with respect to the original learning problem (P-CSL). The size of this neighborhood depends polynomially on $\epsilon_0$, $\epsilon$, the oracle quality $\rho$, and the step size $\eta$. The number of iterations needed to reach this neighborhood is inversely proportional to the desired accuracy $\beta$. It is worth noting that this result applies to the deterministic outputs $(\boldsymbol{\theta}^{(T)}, \boldsymbol{\mu}^{(T)}, \boldsymbol{\lambda}_j^{(T)})$ of Algorithm 1 after convergence and not to a randomized solution obtained by sampling from $(\boldsymbol{\theta}^{(t)}, \boldsymbol{\mu}^{(t)}, \boldsymbol{\lambda}_j^{(t)})$, $t = 0, \ldots, T$ as in [23, 25, 29].

Underlying the oracle in Assumption 4 is often an iterative procedure, e.g., gradient descent, and the cost of running this procedure until convergence to obtain an approximate minimizer can be prohibitive. A common option then is to alternately update the primal variable $\boldsymbol{\theta}^{(t)}$ and the dual variables $(\boldsymbol{\mu}^{(t)}, \boldsymbol{\lambda}_j^{(t)})$. This primal-dual method leads in fact to a classical convex optimization algorithm [69]. While the convergence guarantee of Theorem 3 no longer holds in this case, we observe good results by performing the primal and dual updates at different timescales, e.g., by performing step 3 once per epoch. This is exactly what we do in the next section where we illustrate the usefulness of this constrained learner.

## 6 Numerical experiments

Due to space constraints, we only provide highlights of the results obtained for the problems from Section 3. For more details and additional experiments, see Appendix D in the extended version [56].

**Invariance and fair learning.** In the Adult dataset [70], our goal is to predict whether an individual makes more than US\$ 50,000.00 while being insensitive to gender. If left unconstrained, a small, one-hidden layer NN would change predictions on around 8% of the test samples had their genders been reversed (Fig. 1a). For step 3 of Algorithm 1, we use ADAM [71] with batch size 128 and learning rate 0.1. All other parameters were kept as in the original paper. After each epoch, we update the dual variables (step 4), also using ADAM with a step size of 0.01. All classifiers were trained over 300 epochs.

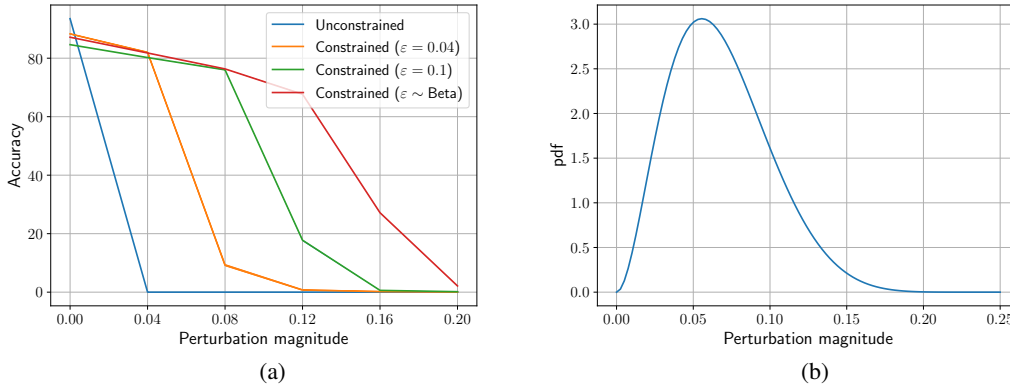

Figure 2: Robust constrained learning (FMNIST): (a) Accuracy of classifiers under the PGD attack for different perturbation magnitudes and (b) distribution of $\varepsilon$ used during training.

When constrained using the pointwise (1), the classifier becomes insensitive to the protected variable in over $99\%$ of the test set. In such simple cases, invariant classifiers can be easily obtained by masking the training samples, although it can bring fairness issues of its own [26, 72]. But Algorithm 1 provides more than an invariant classifier. Due to the bound on the duality gap between (P-CSL) and $(\widehat{\text{D}}\text{-CSL})$, the dual variables have a sensitivity interpretation: the larger their value, the harder the constraint is to satisfy [18]. If we analyze the $20\%$ of individuals with largest $\lambda_n$ (Fig. 1b), we find that a significantly higher prevalence of non-white, non-US natives, married individuals. Clearly, while attempting to control for gender invariance, the constrained learner also had to overcome other prejudices correlated to sexism, a well-known challenge in fair classification [27]. Similar results can be derived when controlling for racial bias in the COMPAS dataset.

**Robust learning.** In this illustration, we use Algorithm 1 to train a ResNet18 [73] to classify images from the FMNIST dataset [74]. As in the previous example, we once again use the ADAM optimizer with the settings from [71]. The best accuracy over the validation set is achieved after 67 epochs, yielding a solution with test accuracy of $93.5\%$ (Figure 2a). However, it fails to classify any of the test images when perturbed using a PGD attack with perturbation magnitude ($\ell_\infty$-norm of the perturbation) as low as $\varepsilon = 0.04$ [30]. The attack uses a step size of $\varepsilon/30$ for 50 iterations and we show the worst result over 10 restarts.

To overcome this issue, we use PGD to sample from a hypothetical "adversarial distribution" $\mathfrak{A}$ and constrain the performance of the solution against $\mathfrak{A}$ as in (PIII). To accelerate training, we use a much weaker attack running PGD without restarts for only 5 steps with step size $\varepsilon/3$. Notice that, as we increase $\varepsilon$, the model becomes increasingly more robust at the cost of nominal performance. Still, the performance degradation remains abrupt. As we argued before, smoother degradation can be obtained by training against a distribution of magnitudes, e.g., the one in Figure 2b. Doing so not only yields better performances under perturbation as well as a small loss of nominal accuracy.

## 7 Conclusion

We put forward a theory of learning under requirements by extending the PAC framework to constrained learning. We then prove that unconstrained and constrained learnability are equivalent by showing that a constrained version of the classical ERM rule is a PACC learner. To overcome the challenges in solving the optimization problem underlying this learner, we derive an alternative learner based on a parametrized empirical dual problem. We show that its approximation error is related to the richness of the parametrization as well as the difficulty of meeting the learning constraint and use it to propose a practical algorithm to learn under requirements. We expect that these generalization results can be used to theoretically ground techniques used in practice to address constrained learning problems beyond fairness and robustness. In particular, similar arguments can be used to develop a constrained theory for reinforcement learning [75]. We also believe that these results can be extended to non-convex losses using recent results on the strong duality of certain non-convex variational problems [68].

## Broader Impact

As learning becomes an ubiquitous technological solution and begins to affect real societal impact, its shortcomings become more evident. A growing number of reports show that its solutions can be prejudiced and prone to tampering or unsafe behaviors [1–6]. Constrained learning allows requirements to be imposed during learning, so that the models and solutions obtained are guaranteed to behave in the desired way despite being learned fully from data. This work provides a framework under which to study learning under requirements and shows how and when it can be done. By providing generalization guarantees on the solutions, it enables learning to be used in critical applications in which there is little tolerance for failure. Naturally, solutions learned under constraints are not necessarily safe or fair. How the learning problem is formulated, i.e., which constraints are imposed, play a definite role on these outcomes and policies determining such requirements can be (and indeed are [76–78]) important sources of biases.

## Acknowledgments and Disclosure of Funding

This work is supported by ARL DCIST CRA W911NF-17-2-0181.

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
