[Supplementary Material]

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

# A    Proof of Theorem 1

Start by noticing from the definition of PACC learnability [more specifically, from (2) in Def. 2] that any PACC learnable class $\mathcal{H}$ is necessarily PAC learnable.

To prove the converse, recall that if $\mathcal{H}$ is PAC learnable, then $\mathcal{H}$ has finite VC dimension [19, Sec. 3.4]. More precisely, for $N > C\zeta^{-1}(\epsilon, \delta, d_{\mathcal{H}})$, where $C$ is an absolute constant and $\zeta^{-1}$ is as in (4), and any bounded function $g$ it holds with probability $1 - \delta$ that

$$\left| \mathbb{E}_{(\boldsymbol{x},y)\sim\mathfrak{D}}\left[ g\big(\phi(\boldsymbol{x}),y\big)\right] - \frac{1}{N}\sum_{n=1}^{N} g\big(\phi(\boldsymbol{x}_n), y_n\big)\right| \leq \epsilon \tag{8}$$

for all function $\phi \in \mathcal{H}$, distributions $\mathcal{D}$, and samples $(\boldsymbol{x}_n, y_n) \sim \mathcal{D}$. Now, let $\hat{\phi}^{\star}$ be a solution of (P-ECRM). From (8) and the boundedness hypothesis on $\ell_0$, we immediately obtain that $\hat{\phi}^{\star}$ is probably approximately optimal as in (2). Additionally, $\hat{\phi}^{\star}$ must be feasible for (P-ECRM). Hence,

$$\frac{1}{N_i}\sum_{n_i=1}^{N_i}\ell_i\big(\hat{\phi}^{\star}(\boldsymbol{x}_{n_i}), y_{n_i}\big) \leq c_i, \qquad \text{for } i = 1,\ldots,m, \quad \text{and} \tag{9a}$$

$$\ell_j\big(\hat{\phi}^{\star}(\boldsymbol{x}_{n_j}), y_{n_j}\big) \leq c_j, \qquad \text{for all } n_j = 1,\ldots,N_j \text{ and } j = m+1,\ldots,m+q. \tag{9b}$$

To show (9) implies that $\hat{\phi}^{\star}$ is a probably approximately feasible, note that we can write, using (8),

$$\mathbb{E}_{(\boldsymbol{x},y)\sim\mathfrak{D}_i}\left[\ell_i\big(\hat{\phi}^{\star}(\boldsymbol{x}),y\big)\right] \leq \frac{1}{N_i}\sum_{n_i=1}^{N_i}\ell_i\big(\hat{\phi}^{\star}(\boldsymbol{x}_{n_i}), y_{n_i}\big) + \epsilon \quad \text{and} \tag{10a}$$

$$\Pr_{(\boldsymbol{x},y)\sim\mathfrak{D}_j}\left[\ell_j\big(\hat{\phi}^{\star}(\boldsymbol{x}),y\big) \leq b_j\right] = \mathbb{E}_{(\boldsymbol{x},y)\sim\mathfrak{D}_j}\left[\mathbb{I}\left[\ell_j\big(\hat{\phi}^{\star}(\boldsymbol{x}),y\big) \leq b_j\right]\right]$$

$$\geq \frac{1}{N_j}\sum_{n_j=1}^{N_j}\mathbb{I}\left[\ell_j\big(\hat{\phi}^{\star}(\boldsymbol{x}_{n_j}), y_{n_j}\big) \leq b_j\right] - \epsilon, \tag{10b}$$

each of which hold with probability $1-\delta$ over the samples $(\boldsymbol{x}_{n_i}, y_{n_i})$ as long as $N_i > C\zeta^{-1}(\epsilon, \delta, d_{\mathcal{H}})$. Combining (9) and (10) we conclude that, with probability $1 - (m+q)\delta$, it holds simultaneously that

$$\mathbb{E}_{(\boldsymbol{x},y)\sim\mathfrak{D}_i}\left[\ell_i\big(\hat{\phi}^{\star}(\boldsymbol{x}),y\big)\right] \leq c_i + \epsilon \quad \text{and}$$

$$\ell_j\big(\hat{\phi}^{\star}(\boldsymbol{x}),y\big) \leq c_j \quad \text{for all } (\boldsymbol{x},y) \in \mathcal{K}_j \subseteq \mathcal{X} \times \mathcal{Y},$$

where each $\mathcal{K}_j$ is a set of $\mathfrak{D}_j$-measure at least $1 - \epsilon$.

Hence, if $\mathcal{H}$ is PAC learnable, then there exists $N$ such that, if $\hat{\phi}^{\star}$ is a solution of (P-ECRM) obtained using $N_i \geq N$ samples from each $\mathfrak{D}_i$, then $\hat{\phi}^{\star}$ is probably approximately optimal as in (2) and probably approximately feasible as in (3). $\square$

# B    Proof of Theorem 2

As we have argued before, we cannot rely on the duality between (PIV) and ($\widehat{\text{D}}$-CSL) to obtain this result because of its non-convexity. Hence, this proof proceeds directly from (P-CSL) by applying three transformations that yield ($\widehat{\text{D}}$-CSL), but whose approximation and estimation errors can be controlled. First, we obtain the dual problem of (P-CSL) and show that this transformation incurs in no error. This stems from the convexity of (P-CSL) under Assumptions 1 and 2 and is a straightforward strong duality result from semi-infinite programming theory (Proposition 1). Second, we approximate the function class $\mathcal{H}$ using the finite dimensional parametrization $f_{\boldsymbol{\theta}}$ and bound the *approximation error* $\epsilon_0$ (Proposition 2). Third, we obtain ($\widehat{\text{D}}$-CSL) by replacing the expectations with their empirical versions. Since the problem is now unconstrained, we can use classical learning theory to evaluate the *estimation error* $\epsilon$ (Proposition 3). We then combine these results to obtain Theorem 2.

Explicitly, we begin by defining the Lagrangian of (P-CSL) as

$$L(\phi, \boldsymbol{\mu}, \boldsymbol{\lambda}) = \mathbb{E}_{(\boldsymbol{x},y)\sim\mathfrak{D}_0}\Big[\ell_0\big(\phi(\boldsymbol{x}),y\big)\Big] + \sum_{i=1}^{m}\mu_i\Big[\mathbb{E}_{(\boldsymbol{x},y)\sim\mathfrak{D}_i}\Big[\ell_i\big(\phi(\boldsymbol{x}),y\big)\Big] - c_i\Big]$$
$$+ \sum_{j=m+1}^{m+n}\int \lambda_j(\boldsymbol{x},y)\Big[\ell_j\big(\phi(\boldsymbol{x}),y\big) - c_j\Big]p_{\mathfrak{D}_j}(\boldsymbol{x},y)d\boldsymbol{x}dy, \tag{11}$$

where $p_{\mathfrak{D}_j}$ is the density of $\mathfrak{D}_j$, $\boldsymbol{\mu} \in \mathbb{R}_+^m$ collects the dual variables $\mu_i$ relative to the expected constraints, and $\boldsymbol{\lambda}$ is an $n \times 1$ vector that collects the functional dual variables $\lambda_j \in L_{1,+}$ relative to the pointwise constraints. By $f \in L_{1,+}$ we mean that $f \in L_1$ (absolutely integrable) and $f \geq 0$ a.e. For conciseness, we leave the measure implicit. Observe that, since the losses $\ell_j$ are bounded (Assumption 1), the integral in (11) exists and is well-defined. This is a direct consequence of Hölder's inequality [78, Thm. 1.5.2]. Additionally, while the result does not require $\mathfrak{D}_j$ to have a density, we assume that it is absolutely continuous with respect to the Lebesgue measure to simplify the derivations. The dual problem of (PV) can then be written as

$$D^\star = \max_{\boldsymbol{\mu}\in\mathbb{R}_+^m,\,\lambda_j\in L_{1,+}}\ \min_{\phi\in\mathcal{H}}\ L(\phi,\boldsymbol{\mu},\boldsymbol{\lambda}). \tag{D-CSL}$$

Assumptions 1–3 imply that (P-CSL) is strongly dual:

**Proposition 1.** *Under Assumptions 1–3, the semi-infinite program* (P-CSL) *and the saddle-point problem* (D-CSL) *are strongly dual, i.e., $P^\star = D^\star$.*

*Proof.* Start by noticing that (P-CSL) can be equivalently formulated as

$$P^\star = \min_{\phi\in\mathcal{H}}\quad \mathbb{E}_{(\boldsymbol{x},y)\sim\mathfrak{D}_0}\Big[\ell_0\big(\phi(\boldsymbol{x}),y\big)\Big]$$
$$\text{subject to}\quad \mathbb{E}_{(\boldsymbol{x},y)\sim\mathfrak{D}_i}\Big[\ell_i\big(\phi(\boldsymbol{x}),y\big)\Big] \leq c_i,\quad i = 1,\dots,m,$$
$$\ell_j\big(\phi(\boldsymbol{x}),y\big)p_{\mathfrak{D}_j}(\boldsymbol{x},y) \leq c_j p_{\mathfrak{D}_j}(\boldsymbol{x},y),\quad (\boldsymbol{x},y) \in \mathcal{X}\times\mathcal{Y}, \tag{PV}$$
$$j = m+1,\dots,n.$$

In fact, both problem have the same objective function and feasibility set. Indeed, if $\mathfrak{D} > 0$, the transformation in the pointwise constraints is vacuous. On the other hand, when $\mathfrak{D}$ vanishes, the constraint is not enforced in (PV). However, neither is it in (P-CSL) since the pointwise constraint need not hold on sets of $\mathfrak{D}$-measure zero. Note that this is different from satisfying the constraint with probability $\mathfrak{D}$.

From Assumptions 1 and 2 we obtain that (PV) is a semi-infinite convex program. What is more, Assumption 3 implies it has a strictly feasible solution $\phi' = f_{\boldsymbol{\theta}'}$. This constraint qualification, sometimes known as *Slater's condition*, implies that is strongly dual, i.e., that $P^\star = D^\star$ [79]. $\qquad\square$

## B.1 The approximation gap

While there is no duality gap between (P-CSL) and (D-CSL), the latter remains a variational problem. The next step is there to approximate the functional space $\mathcal{H}$ by $\mathcal{P} = \{f_{\boldsymbol{\theta}} \mid \boldsymbol{\theta} \in \mathbb{R}^p\}$, the space induced by the finite dimensional parametrization $f_{\boldsymbol{\theta}}$. Thus, (D-CSL) becomes the finite dimensional problem

$$D_\nu^\star = \max_{\boldsymbol{\mu}\in\mathbb{R}_+^m,\,\lambda_j\in L_{1,+}}\ \min_{\boldsymbol{\theta}\in\mathbb{R}^p}\ L_\nu(\boldsymbol{\theta},\boldsymbol{\mu},\boldsymbol{\lambda}) \triangleq L(f_{\boldsymbol{\theta}},\boldsymbol{\mu},\boldsymbol{\lambda}). \tag{$D_\nu$-CSL}$$

Since $\mathcal{P} \subseteq \mathcal{H}$ (Assumption 2), it is clear that $D_\nu^\star \geq D^\star = P^\star$. Yet, if the parametrization is rich enough, we should expect the gap $D_\nu^\star - P^\star$ to be small. This intuition is formalized in the following proposition.

**Proposition 2.** *Let $\boldsymbol{\theta}^\star$ achieve the saddle-point in* ($D_\nu$-CSL). *Under Assumptions 1–3, $f_{\boldsymbol{\theta}^\star}$ is a feasible, near-optimal solution of* (P-CSL). *Explicitly,*

$$P^\star \leq D_\nu^\star \leq P^\star + \left(1 + \|\tilde{\boldsymbol{\mu}}^\star\|_1 + \sum_{j=m+1}^{m+q}\|\tilde{\lambda}_j^\star\|_{L_1}\right)L\nu, \tag{12}$$

*for $P^\star$ and $D_\nu^\star$ defined as in* (P-CSL) *and* ($D_\nu$-CSL) *respectively and where $(\tilde{\boldsymbol{\mu}}^\star, \tilde{\boldsymbol{\lambda}}^\star)$ are the dual variables of* (P-CSL) *with the constraints tightened to $c_i - M\nu$ for $i = 0,\dots,m+q$.*

*Proof.* See Appendix B.4. □

## B.2 The estimation gap

All that remains, is to turn the statistical Lagrangian (11) into the empirical (5). The incurred estimation error is described in the next proposition.

**Proposition 3.** *Let $\hat{\boldsymbol{\theta}}^{\star}$ achieve the saddle-point in* (D̂-CSL) *and for $\delta > 0$, let*

$$\zeta(N) = \sqrt{\frac{1}{N}\left[1 + \log\left(\frac{4(m+q+2)(2N)^{d_{\mathcal{P}}}}{\delta}\right)\right]}, \tag{13}$$

*where $d_{\mathcal{P}}$ is the VC dimension of the parametrized class $\mathcal{P}$. Under Assumptions 1–3, it holds with probability $1 - \delta$ over the samples drawn from the distributions $\mathfrak{D}_i$ that*

$$\left|D_{\nu}^{\star} - \hat{D}^{\star}\right| \leq B\zeta(N_0), \tag{14}$$

$$\mathbb{E}_{(\boldsymbol{x},y)\sim\mathfrak{D}_i}\left[\ell_i\big(f_{\hat{\boldsymbol{\theta}}^{\star}}(\boldsymbol{x}),y\big)\right] \leq c_i + B\zeta(N_i), \text{ and} \tag{15}$$

$$\ell_j\big(f_{\hat{\boldsymbol{\theta}}^{\star}}(\boldsymbol{x}),y\big) \leq c_j \text{ for } (\boldsymbol{x},y)\in\mathcal{K}_j, \tag{16}$$

*where $\mathcal{K}_j \subseteq \mathcal{X}\times\mathcal{Y}$ is a set of $\mathfrak{D}_j$-measure at least $1 - \zeta(N_j)$ for all $j = m+1,\dots,m+q$.*

*Proof.* See appendix B.5. □

## B.3 The PACC solution

The proof concludes by combining the parametrization and estimation gap results from Propositions 2 and 3. Namely, notice that (15) and (16) imply that the minimizer $\hat{\boldsymbol{\theta}}^{\star}$ that achieves the saddle-point in (D̂-CSL) is probably approximately feasible [see (3)] for (P-CSL). Then, combining (12) and (14) using the triangle inequality yields the near-PACC gap from Def. 3. Fixing $N$ such that $B\zeta(N) \leq \epsilon$ yields the result in Theorem 2. □

## B.4 Proof of Proposition 2: The Approximation Gap

We first prove that $f_{\boldsymbol{\theta}^{\star}}$ is feasible for (P-CSL) and then bound the gap between $D_{\nu}^{\star}$ and $P^{\star}$.

**Feasibility.** Suppose that $f_{\boldsymbol{\theta}^{\star}}$ is infeasible. Then, there exists at least one $i > 0$ such that $\mathbb{E}_{(\boldsymbol{x},y)\sim\mathfrak{D}_i}\left[\ell_i\big(f_{\boldsymbol{\theta}^{\star}}(\boldsymbol{x}),y\big)\right] > c_i$ or $\ell_i\big(f_{\boldsymbol{\theta}^{\star}}(\boldsymbol{x}),y\big) > c_i$ over some set $\mathcal{A} \subseteq \mathcal{X}\times\mathcal{Y}$ of positive $\mathfrak{D}_i$-measure. Since $\boldsymbol{\mu}$ and $\boldsymbol{\lambda}$ are unbounded above, we obtain that $D_{\nu}^{\star} \to +\infty$. However, Assumptions 1 and 3 imply that $D_{\nu}^{\star} < +\infty$. Indeed, consider the dual function

$$d(\boldsymbol{\mu},\boldsymbol{\lambda}) = \min_{\boldsymbol{\theta}\in\mathcal{H}} L_{\nu}(\boldsymbol{\theta},\boldsymbol{\mu},\boldsymbol{\lambda})$$

$$= \min_{\boldsymbol{\theta}\in\mathcal{H}} \mathbb{E}_{(\boldsymbol{x},y)\sim\mathfrak{D}_0}\left[\ell_0\big(f_{\boldsymbol{\theta}}(\boldsymbol{x}),y\big)\right] + \sum_{i=1}^{m}\mu_i\left[\mathbb{E}_{(\boldsymbol{x},y)\sim\mathfrak{D}_i}\left[\ell_i\big(f_{\boldsymbol{\theta}}(\boldsymbol{x}),y\big)\right] - c_i\right]$$

$$+ \sum_{j=m+1}^{m+q}\int \lambda_j(\boldsymbol{x},y)\left[\ell_j\big(f_{\boldsymbol{\theta}}(\boldsymbol{x}),y\big) - c_j\right]p_{\mathfrak{D}_j}(\boldsymbol{x},y)d\boldsymbol{x}dy, \tag{17}$$

for the Lagrangian defined in (D$_{\nu}$-CSL). Using the fact that $\ell_0$ is $B$-bounded (Assumption 1) and that there exists a strictly feasible $\boldsymbol{\theta}'$ (Assumption 3), $d(\boldsymbol{\mu},\boldsymbol{\lambda})$ is upper bounded by

$$d(\boldsymbol{\mu},\boldsymbol{\lambda}) \leq \mathbb{E}_{(\boldsymbol{x},y)\sim\mathfrak{D}_0}\left[\ell_0\big(f_{\boldsymbol{\theta}'}(\boldsymbol{x}),y\big)\right] + \sum_{i=1}^{m}\mu_i\left[\mathbb{E}_{(\boldsymbol{x},y)\sim\mathfrak{D}_i}\left[\ell_i\big(f_{\boldsymbol{\theta}\dagger}(\boldsymbol{x}),y\big)\right] - c_i\right]$$

$$+ \sum_{j=m+1}^{m+q}\int \lambda_j(\boldsymbol{x},y)\left[\ell_j\big(f_{\boldsymbol{\theta}'}(\boldsymbol{x}),y\big) - c_j\right]p_{\mathfrak{D}_j}(\boldsymbol{x},y)d\boldsymbol{x}dy < B,$$

where we used the fact that $\mu_i \geq 0$ and $\lambda_j \geq 0$ $\mathfrak{D}_j$-a.e. Hence, it must be that $f_{\boldsymbol{\theta}^{\star}}$ is feasible for (P-CSL).

**Near-optimality.** First, recall that under Assumptions 1–3, (P-CSL)–(D-CSL) form a strongly dual pair of mathematical programs (Proposition 1). For the Lagrangian in (11), we therefore obtain the saddle-point relation

$$L(\phi^\star, \boldsymbol{\mu}', \boldsymbol{\lambda}') \leq \max_{\boldsymbol{\mu}, \boldsymbol{\lambda}} \min_{\phi \in \mathcal{H}} L(\phi, \boldsymbol{\mu}, \boldsymbol{\lambda}) = D^\star = P^\star = \min_{\phi \in \mathcal{H}} \max_{\boldsymbol{\mu}, \boldsymbol{\lambda}} L(\phi, \boldsymbol{\mu}, \boldsymbol{\lambda}) \leq L(\phi', \boldsymbol{\mu}^\star, \boldsymbol{\lambda}^\star)$$

(18)

holds for all $\phi' \in \mathcal{H}$, $\boldsymbol{\mu}' \in \mathbb{R}^m_+$, and $\lambda'_j \in L_{1,+}$, where $\phi^\star$ is a solution of (P-CSL) and $(\boldsymbol{\mu}^\star, \boldsymbol{\lambda}^\star)$ are solutions of (D-CSL). We omit the spaces that $(\boldsymbol{\mu}, \boldsymbol{\lambda})$ belong to for conciseness. Additionally, we have from (D$_\nu$-CSL) that

$$D^\star_\nu \geq \min_{\boldsymbol{\theta} \in \mathbb{R}^p} L(\boldsymbol{\theta}, \boldsymbol{\mu}, \boldsymbol{\lambda}), \quad \text{for all } \boldsymbol{\mu} \in \mathbb{R}^m_+ \text{ and } \lambda_j \in L_{1,+}.$$

(19)

Immediately, we obtain the lower bound in (12). Explicitly,

$$D^\star_\nu \geq \min_{\boldsymbol{\theta} \in \mathbb{R}^p} L(\boldsymbol{\theta}, \boldsymbol{\mu}, \boldsymbol{\lambda}) \geq \min_{\phi \in \mathcal{H}} L(\phi, \boldsymbol{\mu}^\star, \boldsymbol{\lambda}^\star) = P^\star,$$

(20)

where the second inequality comes from the fact that $\mathcal{P} \subseteq \mathcal{H}$ (Assumption 2).

The upper bound is obtained by relating the parameterized dual problem (D$_\nu$-CSL) to a perturbed (tightened) version of the original (P-CSL). To do so, start by adding and subtracting $L(\phi, \boldsymbol{\mu}, \boldsymbol{\lambda})$ from (D$_\nu$-CSL) to get

$$D^\star_\nu = \max_{\boldsymbol{\mu}, \boldsymbol{\lambda}} \min_{\boldsymbol{\theta} \in \mathbb{R}^p} L(\phi, \boldsymbol{\mu}, \boldsymbol{\lambda}) + \mathbb{E}_{(\boldsymbol{x},y) \sim \mathfrak{D}_0} \Big[ \ell_0\big(f_{\boldsymbol{\theta}}(\boldsymbol{x}), y\big) - \ell_0(\phi(\boldsymbol{x}), y) \Big]$$

$$+ \sum_{i=1}^m \mu_i \, \mathbb{E}_{(\boldsymbol{x},y) \sim \mathfrak{D}_i} \Big[ \ell_i\big(f_{\boldsymbol{\theta}}(\boldsymbol{x}), y\big) - \ell_i(\phi(\boldsymbol{x}), y) \Big]$$

$$+ \sum_{j=m+1}^{m+q} \mathbb{E}_{(\boldsymbol{x},y) \sim \mathfrak{D}_j} \Big[ \lambda_j(\boldsymbol{x}, y) \big( \ell_j\big(f_{\boldsymbol{\theta}}(\boldsymbol{x}), y\big) - \ell_j(\phi(\boldsymbol{x}), y) \big) \Big],$$

(21)

where we wrote the integral against $p_{\mathfrak{D}_j}$ as an expectation for conciseness. Then, using the fact that $\ell_i$ is $M$-Lipschitz continuous (Assumption 1), we bound the expectations in the first two terms of (21) as

$$\mathbb{E}_{(\boldsymbol{x},y) \sim \mathfrak{D}_j} \Big[ \ell_i\big(f_{\boldsymbol{\theta}}(\boldsymbol{x}), y\big) - \ell_i(\phi(\boldsymbol{x}), y) \Big] \leq \mathbb{E}_{(\boldsymbol{x},y) \sim \mathfrak{D}_j} \Big[ \big| \ell_i\big(f_{\boldsymbol{\theta}}(\boldsymbol{x}), y\big) - \ell_i(\phi(\boldsymbol{x}), y) \big| \Big]$$

$$\leq M \, \mathbb{E}_{(\boldsymbol{x},y) \sim \mathfrak{D}_j} \Big[ \big| f_{\boldsymbol{\theta}}(\boldsymbol{x}) - \phi(\boldsymbol{x}) \big| \Big], \text{ for } i = 0, \ldots, m.$$

(22)

To bound the last expectation in (21), we first use Hölder's inequality to get

$$\mathbb{E}_{(\boldsymbol{x},y) \sim \mathfrak{D}_j} \Big[ \lambda_j(\boldsymbol{x}, y) \big( \ell_j\big(f_{\boldsymbol{\theta}}(\boldsymbol{x}), y\big) - \ell_j(\phi(\boldsymbol{x}), y) \big) \Big] \leq$$

$$\mathbb{E}_{(\boldsymbol{x},y) \sim \mathfrak{D}_j} \big[ \lambda_j(\boldsymbol{x}, y) \big] \, \big\| \ell_j\big(f_{\boldsymbol{\theta}}(\boldsymbol{x}), y\big) - \ell_j(\phi(\boldsymbol{x}), y) \big\|_{L_\infty},$$

where we recall that $\|g\|_{L_\infty}$ is the essential supremum of $|g|$. Then, the $M$-Lipschitz continuity of $\ell_j$ (Assumption 1) implies that

$$\mathbb{E}_{(\boldsymbol{x},y) \sim \mathfrak{D}_j} \Big[ \lambda_j(\boldsymbol{x}, y) \big( \ell_j\big(f_{\boldsymbol{\theta}}(\boldsymbol{x}), y\big) - \ell_j(\phi(\boldsymbol{x}), y) \big) \Big] \leq$$

$$M \, \| f_{\boldsymbol{\theta}}(\boldsymbol{x}) - \phi(\boldsymbol{x}) \|_{L_\infty} \, \mathbb{E}_{(\boldsymbol{x},y) \sim \mathfrak{D}_j} \big[ \lambda_j(\boldsymbol{x}, y) \big]. \quad (23)$$

Using (22) and (23), together with the approximation property of the class $\mathcal{H}$ (Assumption 2), we upper bound the minimum over $\boldsymbol{\theta}$ in (21) to obtain

$$D^\star_\nu \leq \max_{\boldsymbol{\mu}, \boldsymbol{\lambda}} L(\phi, \boldsymbol{\mu}, \boldsymbol{\lambda}) + \left[ 1 + \sum_{i=1}^m \mu_i + \sum_{j=m+1}^{m+q} \mathbb{E}_{(\boldsymbol{x},y) \sim \mathfrak{D}_j} \big[ \lambda_j(\boldsymbol{x}, y) \big] \right] M\nu.$$

(24)

Notice that since (24) holds uniformly for all $\phi \in \mathcal{H}$, it also holds for the minimizer

$$D^\star_\nu \leq \min_{\phi \in \mathcal{H}} \max_{\boldsymbol{\mu}, \boldsymbol{\lambda}} L(\phi, \boldsymbol{\mu}, \boldsymbol{\lambda}) + \left[ 1 + \sum_{i=1}^m \mu_i + \sum_{j=m+1}^{m+n} \mathbb{E}_{(\boldsymbol{x},y) \sim \mathfrak{D}_j} \big[ \lambda_j(\boldsymbol{x}, y) \big] \right] M\nu \triangleq \tilde{P}^\star$$

(25)

and that the right-hand side of (25), namely $\tilde{P}^\star$, is in fact a perturbed version of (P-CSL). Hence, we obtain another saddle-point relation similar to (18) relating $\tilde{P}^\star$, and consequently $D_\nu^\star$, to $P^\star$.

Formally, (25) can be rearranged as

$$
\begin{aligned}
\tilde{P}^\star = \min_{\phi \in \mathcal{H}} \max_{\boldsymbol{\mu}, \boldsymbol{\lambda}} \ & \mathbb{E}_{(\boldsymbol{x},y) \sim \mathfrak{D}_0} \Big[ \ell_0\big(\phi(\boldsymbol{x}), y\big) + M\nu \Big] \\
& + \sum_{i=1}^{m} \mu_i \left[ \mathbb{E}_{(\boldsymbol{x},y) \sim \mathfrak{D}_i} \Big[ \ell_i\big(\phi(\boldsymbol{x}), y\big) \Big] - c_i + M\nu \right] \\
& + \sum_{j=m+1}^{m+q} \int \lambda_j(\boldsymbol{x}, y) \Big[ \ell_j\big(\phi(\boldsymbol{x}), y\big) - c_j + M\nu \Big] p_{\mathfrak{D}_j}(\boldsymbol{x}, y) d\boldsymbol{x} dy,
\end{aligned}
\tag{26}
$$

where we recognize the optimization problem of

$$
\begin{aligned}
\tilde{P}^\star = \min_{\phi \in \mathcal{H}} \quad & \mathbb{E}_{(\boldsymbol{x},y) \sim \mathfrak{D}_0} \Big[ \ell_0\big(\phi(\boldsymbol{x}), y\big) \Big] + M\nu \\
\text{subject to} \quad & \mathbb{E}_{(\boldsymbol{x},y) \sim \mathfrak{D}_i} \Big[ \ell_i\big(\phi(\boldsymbol{x}), y\big) \Big] \leq c_i - M\nu, \quad i = 1, \dots, m, \\
& \ell_j\big(\phi(\boldsymbol{x}), y\big) \leq c_j - M\nu \quad \mathfrak{D}_j\text{-a.e.}, \qquad j = m+1, \dots, m+q.
\end{aligned}
\tag{PVI}
$$

Under Assumptions 1–3, (PVI) is also strongly dual (Proposition 1), so that

$$
\tilde{P}^\star = \min_{\phi \in \mathcal{H}} L(\phi, \tilde{\boldsymbol{\mu}}^\star, \tilde{\boldsymbol{\lambda}}^\star) + \left[ 1 + \sum_{i=1}^{m} \tilde{\mu}_i^\star + \sum_{j=m+1}^{m+n} \mathbb{E}_{(\boldsymbol{x},y) \sim \mathfrak{D}_j} \big[ \tilde{\lambda}_j^\star(\boldsymbol{x}, y) \big] \right] M\nu,
\tag{27}
$$

where $(\tilde{\boldsymbol{\mu}}^\star, \tilde{\boldsymbol{\lambda}}^\star)$ are the dual variables of (PVI), i.e., the $(\boldsymbol{\mu}, \boldsymbol{\lambda})$ that achieve

$$
\begin{aligned}
\tilde{D}^\star = \max_{\boldsymbol{\mu}, \boldsymbol{\lambda}} \min_{\phi \in \mathcal{H}} \ & \mathbb{E}_{(\boldsymbol{x},y) \sim \mathfrak{D}_0} \Big[ \ell_0\big(\phi(\boldsymbol{x}), y\big) + M\nu \Big] \\
& + \sum_{i=1}^{m} \mu_i \left[ \mathbb{E}_{(\boldsymbol{x},y) \sim \mathfrak{D}_i} \Big[ \ell_i\big(\phi(\boldsymbol{x}), y\big) \Big] - c_i + M\nu \right] \\
& + \sum_{j=m+1}^{m+q} \int \lambda_j(\boldsymbol{x}, y) \Big[ \ell_j\big(\phi(\boldsymbol{x}), y\big) - c_j + M\nu \Big] p_{\mathfrak{D}_j}(\boldsymbol{x}, y) d\boldsymbol{x} dy.
\end{aligned}
\tag{28}
$$

Going back to (25) we can now conclude the proof. First, use (27) to obtain

$$
D_\nu^\star \leq \tilde{P}^\star \leq L(\phi^\star, \tilde{\boldsymbol{\mu}}^\star, \tilde{\boldsymbol{\lambda}}^\star) + \left[ 1 + \|\tilde{\boldsymbol{\mu}}^\star\|_1 + \sum_{j=m+1}^{m+q} \left\| \tilde{\lambda}_j^\star \right\|_{L_1} \right] L\nu,
\tag{29}
$$

where we used $\phi^\star$, the solution of (P-CSL), as a suboptimal solution in (27) and exploited the fact that the dual variables are non-negative to write their sum (integral) as an $\ell_1$-norm ($L_1$-norm). The saddle point relation (18) gives $L(\phi^\star, \tilde{\boldsymbol{\mu}}^\star, \tilde{\boldsymbol{\lambda}}^\star) \leq P^\star$, from which we obtain the desired upper bound in (12). $\qquad\square$

## B.5 Proof of Proposition 3: The Estimation Gap

**Feasibility.** The proof follows by first showing that $\hat{\theta}^\star$ must be feasible for the parametrized ECRM (PIV) using the same argument as in Sec. (B.4). We then proceed as in the proof of Theorem 1.

Formally, suppose there exists at least one $i > 0$ such that

$$
\frac{1}{N_i} \sum_{n_i=1}^{N_i} \ell_i\big(f_{\hat{\boldsymbol{\theta}}^\star}(\boldsymbol{x}_{n_i}), y_{n_i}\big) > c_i \quad \text{or} \quad \ell_i\big(f_{\hat{\boldsymbol{\theta}}^\star}(\boldsymbol{x}_{n_i}), y_{n_i}\big) > c_i \text{ for some } n_i.
$$

Then, since $\boldsymbol{\mu}$ and $\boldsymbol{\lambda}_j$ are unbounded above, we obtain that $\hat{D}^\star \to +\infty$. However, Assumptions 1 and 3 imply that $\hat{D}^\star < +\infty$. Indeed, consider the empirical dual function

$$
\hat{d}(\boldsymbol{\mu}, \boldsymbol{\lambda}_j) = \min_{\boldsymbol{\theta} \in \mathbb{R}^p} \hat{L}(\boldsymbol{\theta}, \boldsymbol{\mu}, \boldsymbol{\lambda}_j).
\tag{30}
$$

Using the fact that $\ell_0$ is $B$-bounded (Assumption 1) and that there exists a strictly feasible $\boldsymbol{\theta}^\dagger$ (Assumption 3), $\hat{d}(\boldsymbol{\mu}, \boldsymbol{\lambda}) < B$. Hence, it must be that

$$\frac{1}{N_i} \sum_{n_i=1}^{N_i} \ell_i\big(f_{\hat{\boldsymbol{\theta}}^\star}(\boldsymbol{x}_{n_i}), y_{n_i}\big) \le c_i, \qquad \text{for } i = 1, \ldots, m, \quad \text{and} \tag{31a}$$

$$\ell_j\big(f_{\hat{\boldsymbol{\theta}}^\star}(\boldsymbol{x}_{n_j}), y_{n_j}\big) \le c_j, \qquad \text{for all } n_j \text{ and } j = m+1, \ldots, m+q. \tag{31b}$$

We now proceed to use the classic VC bound [19, Sec. 3.4] to show that $f_{\hat{\boldsymbol{\theta}}^\star}$ is a probably approximately feasible solution of (P-CSL). To do so, recall from (8) that since the $\ell_i$ are bounded (Assumption 1) and $\mathcal{P}$ has finite VC dimension $d_{\mathcal{P}}$, we obtain that

$$\mathbb{E}_{(\boldsymbol{x},y)\sim\mathfrak{D}_i}\Big[\ell_i\big(f_{\boldsymbol{\theta}}(\boldsymbol{x}), y\big)\Big] \le \frac{1}{N_i} \sum_{n_i=1}^{N_i} \ell_i\big(f_{\boldsymbol{\theta}}(\boldsymbol{x}_{n_i}), y_{n_i}\big) + B\zeta(N_i) \quad \text{and} \tag{32a}$$

$$\Pr_{(\boldsymbol{x},y)\sim\mathfrak{D}_j}\Big[\ell_j\big(f_{\boldsymbol{\theta}}(\boldsymbol{x}), y\big) \le b_j\Big] = \mathbb{E}_{(\boldsymbol{x},y)\sim\mathfrak{D}_j}\Big[\mathbb{I}\big[\ell_j\big(f_{\boldsymbol{\theta}}(\boldsymbol{x}), y\big) \le b_j\big]\Big]$$

$$\ge \frac{1}{N_j} \sum_{n_j=1}^{N_j} \mathbb{I}\big[\ell_j\big(f_{\boldsymbol{\theta}}(\boldsymbol{x}_{n_j}), y_{n_j}\big) \le b_j\big] - \zeta(N_j) \tag{32b}$$

hold with probability $1 - \delta$ over the datasets $\{(\boldsymbol{x}_{n_i}), y_{n_i})\}_i$ for $\zeta$ as in (13). Combining (31) and (32) and using the union bound, we conclude that, with probability $1 - (m+q)\delta$,

$$\mathbb{E}_{(\boldsymbol{x},y)\sim\mathfrak{D}_i}\Big[\ell_i\big(f_{\hat{\boldsymbol{\theta}}^\star}(\boldsymbol{x}), y\big)\Big] \le b_i + B\zeta(N_i) \quad \text{and}$$

$$\ell_j\big(f_{\hat{\boldsymbol{\theta}}^\star}(\boldsymbol{x}), y\big) \le b_j \quad \text{for all } (\boldsymbol{x}, y) \in \mathcal{K}_j \subseteq \mathcal{X} \times \mathcal{Y},$$

where $\mathcal{K}_j$ is a set of $\mathfrak{D}_j$-measure at least $1 - \zeta(N_j)$.

**Near-optimality.** Let $(\boldsymbol{\theta}_\nu^\star, \boldsymbol{\mu}_\nu^\star, \boldsymbol{\lambda}_\nu^\star)$ and $(\hat{\boldsymbol{\theta}}^\star, \hat{\boldsymbol{\mu}}^\star, \hat{\boldsymbol{\lambda}}^\star)$ be variables that achieve $D_\nu^\star$ in (D$_\nu$-CSL) and $\hat{D}^\star$ in ($\hat{\text{D}}$-CSL) respectively. Then, it holds that

$$\mu_{\nu,j}^\star\Big(\mathbb{E}\left[\ell_i(f(\boldsymbol{\theta}_\nu^\star, \boldsymbol{x}), y)\right] - c_j\Big) = 0, \tag{33a}$$

$$\lambda_{\nu,j}^\star(\boldsymbol{x}, y)\Big(\ell_j(f(\boldsymbol{\theta}_\nu^\star, \boldsymbol{x}), y) - c_j\Big) = 0, \quad \mathfrak{D}_j\text{-a.e.}, \tag{33b}$$

$$\hat{\mu}_i\Big(\frac{1}{N} \sum_{n=1}^N \ell_i(f(\hat{\boldsymbol{\theta}}^\star, \boldsymbol{x}_n), y_n) - c_i\Big) = 0, \quad \text{and} \tag{33c}$$

$$\hat{\lambda}_{j,n_j}\Big(\ell_i(f(\hat{\boldsymbol{\theta}}^\star, \boldsymbol{x}_n), y_n) - c_j\Big) = 0, \tag{33d}$$

known as *complementary slackness* conditions. While these are part of the classical KKT conditions [66, Sec. 5.5.3], it should be noted that the non-convex nature of both (D$_\nu$-CSL) and ($\hat{\text{D}}$-CSL) implies that these are only necessary and not sufficient for optimality. Nevertheless, feasibility is enough to establish (33).

Indeed, recall from Proposition 2 and (31) that the constraint slacks in parentheses in (33) are non-positive. Hence, the left-hand sides in (33) are also non-positive and if (33a) does not hold for some $i$ or if (33b) does not hold for some $j$ and a set $\mathcal{Z}_j$ of positive $\mathfrak{D}_j$ measure, then letting $\mu_{\nu,i}^\star = 0$ or making $\lambda_j(\boldsymbol{x}, y)$ vanish over $\mathcal{Z}_j$ would increase the value of $D_\nu^\star$, contradicting its optimality. Note that since $\mathcal{Z}_j$ is measurable, the modified $\lambda_j$ would still be measurable. A similar argument applies to (33c) and (33d).

Immediately, (33) implies that both (D$_\nu$-CSL) and ($\hat{\text{D}}$-CSL) reduce to

$$D_\nu^\star = \mathbb{E}\left[\ell_0\left(f(\boldsymbol{\theta}_\nu^\star, \boldsymbol{x}), y\right)\right] \qquad \triangleq F_0(\boldsymbol{\theta}_\nu^\star) \quad \text{and} \tag{34a}$$

$$\hat{D}^\star = \frac{1}{N_0} \sum_{n_0=1}^{N_0} \ell_0\left(f(\hat{\boldsymbol{\theta}}^\star, \boldsymbol{x}_{n_0}), y_{n_0}\right) \triangleq \hat{F}_0(\hat{\boldsymbol{\theta}}^\star). \tag{34b}$$

To proceed, use the optimality of $\boldsymbol{\theta}_\nu^\star$ and $\boldsymbol{\theta}$ for $F_0$ and $\hat{F}_0$ respectively to write

$$F_0(\boldsymbol{\theta}_\nu^\star) - \hat{F}_0(\boldsymbol{\theta}_\nu^\star) \leq F_0(\boldsymbol{\theta}_\nu^\star) - \hat{F}_0(\hat{\boldsymbol{\theta}}^\star) \leq F_0(\hat{\boldsymbol{\theta}}^\star) - \hat{F}_0(\hat{\boldsymbol{\theta}}^\star).$$

Then, (34) yields the bound

$$\left| D_\nu^\star - \hat{D}^\star \right| = \left| F_0(\boldsymbol{\theta}_\nu^\star) - \hat{F}_0(\hat{\boldsymbol{\theta}}^\star) \right| \leq \max \left\{ \left| F_0(\boldsymbol{\theta}_\nu^\star) - \hat{F}_0(\boldsymbol{\theta}_\nu^\star) \right|, \left| F_0(\hat{\boldsymbol{\theta}}^\star) - \hat{F}_0(\hat{\boldsymbol{\theta}}^\star) \right| \right\} \tag{35}$$

and applying the VC generalization bound from [19, Sec. 3.4] to (35), yields that, uniformly over $\boldsymbol{\theta}$,

$$\left| F_0(\boldsymbol{\theta}) - \hat{F}_0(\boldsymbol{\theta}) \right| \leq B\zeta(N_0), \tag{36}$$

with probability $1 - \delta$ and for $\zeta$ as in (4). Combining (35) and (36) concludes the proof. $\qquad\square$

## C   Proof of Theorem 3

In this appendix, we prove the following quantitative version of Theorem 3:

**Theorem 4.** *Fix $\beta > 0$ and consider Algorithm 1 with at least $C\zeta^{-1}(\epsilon, \delta, d_{\mathcal{P}})$ samples from each $\mathfrak{D}_j$, where $C$ is an absolute constant, $\zeta^{-1}$ is as in (4), and $d_{\mathcal{P}}$ is the VC dimension of $\mathcal{P}$. Under Assumptions 1–4, Algorithm 1 converges to a probably approximately feasible solution and*

$$P^\star - \rho - \frac{\eta}{2}S - \beta - \epsilon \leq \hat{L}\left(\boldsymbol{\theta}^{(T)}, \boldsymbol{\mu}^{(T)}, \boldsymbol{\lambda}^{(T)}\right) \leq P^\star + \rho + \epsilon_0 + \epsilon \tag{37}$$

*with probability $1 - \delta$ after $T$ steps for $\epsilon_0$ as in (6),*

$$S = \sum_{i=1}^{m} (B - c_i)^2 + \sum_{j=m+1}^{m+q} \frac{1}{N_j}(B - c_j)^2, \tag{38}$$

*and*

$$T \leq \frac{U_0}{2\eta\beta} + 1,$$

*where $U_0$ is the distance to a pair of optimal dual variables at the beginning of the algorithm, namely,*

$$U_0 = \|\boldsymbol{\mu}^\star\|^2 + \sum_{j=m+1}^{m+q} \left\|\boldsymbol{\lambda}_j^\star\right\|^2 \tag{39}$$

*for $(\boldsymbol{\mu}^\star, \boldsymbol{\lambda}_j^\star)$ solutions of $(\widehat{\text{D}}\text{-CSL})$.*

**Near-optimality.**   We proceed by proving that

$$\hat{D}^\star - \rho - \frac{\eta}{2}S - \beta \leq \hat{L}\left(\boldsymbol{\theta}^{(T)}, \boldsymbol{\mu}^{(T)}, \boldsymbol{\lambda}^{(T)}\right) \leq \hat{D}^\star + \rho, \tag{40}$$

from which we obtain (37) by recalling that $\hat{D}^\star$ is near-PACC (Theorem 2). More precisely, by using Propositions 2 and 3.

Start by defining the empirical dual function

$$\hat{d}(\boldsymbol{\mu}, \boldsymbol{\lambda}_j) \triangleq \min_{\boldsymbol{\theta}} \hat{L}(\boldsymbol{\theta}, \boldsymbol{\mu}, \boldsymbol{\lambda}_j). \tag{41}$$

The upper bound in (40) then holds trivially from the fact that $\hat{d}(\boldsymbol{\mu}, \boldsymbol{\lambda}_j) \leq \hat{D}^\star$ for all $(\boldsymbol{\mu}, \boldsymbol{\lambda}_j)$. Then, from the characteristics of the approximate minimizer $\boldsymbol{\theta}^{(t)} = \boldsymbol{\theta}^\dagger(\boldsymbol{\mu}^{(t)}, \boldsymbol{\lambda}^{(t)})$ in Assumption 4 we obtain that

$$\hat{L}\left(\boldsymbol{\theta}^{(t)}, \boldsymbol{\mu}^{(t)}, \boldsymbol{\lambda}^{(t)}\right) \leq \hat{D}^\star + \rho, \quad \text{for all } t \geq 0. \tag{42}$$

For the lower bound, we rely on the following relaxation of Dankin's classical theorem [51, Ch. 3]:

**Lemma 1.** *Let $\boldsymbol{\theta}^{\dagger}$ be the approximate minimizer of the empirical Lagrangian (5) at $(\boldsymbol{\mu}, \boldsymbol{\lambda}_j)$ from Assumption 4. Then, the constraint slacks are* approximate *subgradients of the dual function (41), i.e.,*

$$\hat{d}(\boldsymbol{\mu}, \boldsymbol{\lambda}_j) \geq \hat{d}(\boldsymbol{\mu}', \boldsymbol{\lambda}_j') + \sum_{i=1}^{m} (\mu_i - \mu_i') \left[ \frac{1}{N_i} \sum_{n_i=1}^{N_i} \ell_i\big(f_{\boldsymbol{\theta}^{\dagger}}(\boldsymbol{x}_{n_i}), y_{n_i}\big) - c_i \right]$$

$$+ \sum_{j=m+1}^{m+q} \left[ \frac{1}{N_j} \sum_{n_j=1}^{N_j} \big(\lambda_{j,n_j} - \lambda_{j,n_j}'\big) \big(\ell_j\big(f_{\boldsymbol{\theta}^{\dagger}}(\boldsymbol{x}_{n_j}), y_{n_j}\big) - c_j\big) \right] - \rho \tag{43}$$

*for all $(\boldsymbol{\mu}', \boldsymbol{\lambda}_j')$.*

*Proof.* From Assumption 4, we obtain that

$$d(\boldsymbol{\mu}', \boldsymbol{\lambda}_j') \leq d(\boldsymbol{\mu}', \boldsymbol{\lambda}_j') + d(\boldsymbol{\mu}, \boldsymbol{\lambda}_j) - \hat{L}\big(\boldsymbol{\theta}^{\dagger}(\boldsymbol{\mu}\boldsymbol{\lambda}_j), \boldsymbol{\mu}, \boldsymbol{\lambda}_j\big) + \rho. \tag{44}$$

Additionally, we can upper bound (44) by replacing the optimal minimizer in $d(\boldsymbol{\mu}', \boldsymbol{\lambda}_j')$ by any $\boldsymbol{\theta}$. In particular, we can choose $\boldsymbol{\theta}^{\dagger}(\boldsymbol{\mu}, \boldsymbol{\lambda}_j)$ to get

$$d(\boldsymbol{\mu}', \boldsymbol{\lambda}_j') \leq \hat{L}\big(\boldsymbol{\theta}^{\dagger}(\boldsymbol{\mu}, \boldsymbol{\lambda}_j), \boldsymbol{\mu}', \boldsymbol{\lambda}_j'\big) + d(\boldsymbol{\mu}, \boldsymbol{\lambda}_j) - \hat{L}\big(\boldsymbol{\theta}^{\dagger}(\boldsymbol{\mu}, \boldsymbol{\lambda}_j), \boldsymbol{\mu}, \boldsymbol{\lambda}_j\big) + \rho. \tag{45}$$

Notice from (5) that the first term of the Lagrangians in (45) are identical. By expanding them, (45) can then be rearranged as in (43). □

To proceed, let $(\boldsymbol{\mu}^{\star}, \boldsymbol{\lambda}_j^{\star})$ be solutions of the dual problem $(\widehat{\text{D-CSL}})$. We show next that for at least $T = O(1/\beta)$, the total distance

$$U_t = \left\| \boldsymbol{\mu}^{(t)} - \boldsymbol{\mu}^{\star} \right\|^2 + \sum_{j=m+1}^{m+q} \left\| \boldsymbol{\lambda}_j^{(t)} - \boldsymbol{\lambda}^{\star} \right\|^2 \tag{46}$$

decreases by at least $O(\beta)$. To do so, use the updates from Algorithm 1 to write (46) as

$$U_t = \sum_{i=1}^{m} \left\{ \left[ \mu_i^{(t-1)} + \eta \left( \frac{1}{N_i} \sum_{n_i=1}^{N_i} \ell_i\big(f_{\boldsymbol{\theta}^{(t-1)}}(\boldsymbol{x}_{n_i}), y_{n_i}\big) - c_i \right) \right]_+ - \mu_i^{\star} \right\}^2$$

$$+ \sum_{j=m+1}^{m+q} \sum_{n_j=1}^{N_j} \left\{ \left[ \lambda_{j,n_j}^{(t-1)} + \frac{\eta}{N_j} \big(\ell_j\big(f_{\boldsymbol{\theta}^{(t-1)}}(\boldsymbol{x}_{n_j}), y_{n_j}\big) - c_j\big) \right]_+ - \lambda_{j,n_j}^{\star} \right\}^2.$$

Since both $\boldsymbol{\mu}^{\star}$ and $\boldsymbol{\lambda}^{\star}$ belong to the non-negative orthant, we can then use the non-expansiveness of the projection $[\cdot]_+$ [18] to obtain

$$U_t = \sum_{i=1}^{m} \left[ \mu_i^{(t-1)} + \eta \left( \frac{1}{N_i} \sum_{n_i=1}^{N_i} \ell_i\big(f_{\boldsymbol{\theta}^{(t-1)}}(\boldsymbol{x}_{n_i}), y_{n_i}\big) - c_i \right) - \mu_i^{\star} \right]^2$$

$$+ \sum_{j=m+1}^{m+q} \sum_{n_j=1}^{N_j} \left[ \lambda_{j,n_j}^{(t-1)} + \frac{\eta}{N_j} \big(\ell_j\big(f_{\boldsymbol{\theta}^{(t-1)}}(\boldsymbol{x}_{n_j}), y_{n_j}\big) - c_j\big) - \lambda_{j,n_j}^{\star} \right]^2. \tag{47}$$

By expanding the norms in (47), we get that

$$U_t \leq U_{t-1} + 2\eta \left[ \sum_i \big(\mu_i^{(t-1)} - \mu_i^{\star}\big) \left( \frac{1}{N_i} \sum_{n_i=1}^{N_i} \ell_i\big(f_{\boldsymbol{\theta}^{(t-1)}}(\boldsymbol{x}_{n_i}), y_{n_i}\big) - c_i \right) \right.$$

$$\left. + \sum_j \sum_{n_j=1}^{N_j} \frac{1}{N_j} \big(\lambda_{j,n_j}^{(t-1)} - \lambda_{j,n_j}^{\star}\big) \big(\ell_j\big(f_{\boldsymbol{\theta}^{(t-1)}}(\boldsymbol{x}_{n_j}), y_{n_j}\big) - c_j\big) \right]$$

$$+ \eta^2 \left[ \sum_{i=1}^{m} \left[ \frac{1}{N_i} \sum_{n_i=1}^{N_i} \ell_i\big(f_{\boldsymbol{\theta}^{(t-1)}}(\boldsymbol{x}_{n_i}), y_{n_i}\big) - c_i \right]^2 + \sum_{j=m+1}^{m+q} \sum_{n_j=1}^{N_j} \frac{1}{N_j^2} \big[\ell_j\big(f_{\boldsymbol{\theta}^{(t-1)}}(\boldsymbol{x}_{n_j}), y_{n_j}\big) - c_j\big]^2 \right]. \tag{48}$$

Using the fact that the $\ell_i$ are bounded (Assumption 1), the last term in (48) is upper bounded by

$$S = \sum_{i=1}^{m}(B - c_i)^2 + \sum_{j=m+1}^{m+q} \frac{1}{N_j}(B - c_j)^2 = O(B^2).$$

What is more, Lemma 1 can be used to bound the second term in (48) and write

$$U_t \le U_{t-1} + 2\eta\left[\hat{d}\left(\boldsymbol{\mu}^{(t-1)}, \boldsymbol{\lambda}_j^{(t-1)}\right) - \hat{D}^\star + \rho\right] + \eta^2 S,$$

where we used the fact that $\hat{D}^\star = \hat{d}(\boldsymbol{\mu}^\star, \boldsymbol{\lambda}_j^\star)$. Solving the recursion then yields

$$U_t \le U_0 + 2\eta\sum_{t=0}^{t-1}\Delta_t, \tag{49}$$

for

$$\Delta_t = \hat{d}\left(\boldsymbol{\mu}^{(t-1)}, \boldsymbol{\lambda}_j^{(t-1)}\right) - \hat{D}^\star + \rho + \frac{\eta}{2}S. \tag{50}$$

To conclude, notice that $\hat{d}(\boldsymbol{\mu}, \boldsymbol{\lambda}_j) \le \hat{D}^\star$ for all $(\boldsymbol{\mu}, \boldsymbol{\lambda}_j)$. Hence, when $\boldsymbol{\mu}^{(t)}$ and $\boldsymbol{\lambda}_j^{(t)}$ are sufficiently far from the optimum and the step size $\eta$ is sufficiently small, we have $\Delta_t \le 0$ and (49) shows that the distance to the optimum $U_t$ decreases. Formally, fix a precision $\beta > 0$ and let $T = \min\{t \mid \Delta_t > -\beta\}$. Then, from the definition of $\Delta_t$ we obtain the desired lower bound

$$\Delta_T > -\beta \Leftrightarrow \hat{d}\left(\boldsymbol{\mu}^{(t-1)}, \boldsymbol{\lambda}_j^{(t-1)}\right) > \hat{D}^\star - \rho - \frac{\eta}{2}S - \beta$$

What is more, (49) yields

$$T \le \frac{U_0}{2\eta\beta} + 1 = O(\beta^{-1}).$$

$\square$

# D Numerical experiments: additional details

## D.1 Invariance and fair learning

We begin with our analysis of the Adult dataset [69], in which our goal is to predict whether an individual makes more than US\$ 50,000.00 while being insensitive to gender. The transformations performed on the data are listed in Table D.1. We use a neural network with two outputs and a single hidden-layer with 64 nodes using a sigmoidal activation function. The output is encoded into a probability using a softmax transformation ($f_{\boldsymbol{\theta}} : \mathcal{X} \to [0, 1]^2$). Using this parametrization, we then pose the constrained learning problem

$$\begin{aligned} \underset{\boldsymbol{\theta}\in\mathbb{R}^p}{\text{minimize}} \quad & \mathbb{E}\left[\ell_0\left(f_{\boldsymbol{\theta}}(\boldsymbol{x}), y\right)\right] \\ \text{subject to} \quad & \mathrm{D_{KL}}\left(f_{\boldsymbol{\theta}}(\boldsymbol{x}, z) \,\|\, f_{\boldsymbol{\theta}}(\boldsymbol{x}, 1 - z)\right) \le c, \end{aligned} \tag{PVII}$$

where $z$ is the variable gender (encoded 0 for female and 1 for male) and $\ell_0$ is the negative logistic log-likelihood, i.e., $-\log\left([f_{\boldsymbol{\theta}}(\boldsymbol{x})]_y\right)$. To solve (PVII), we use ADAM [70] for step 3 of Algorithm 1, with batch size 128 and learning rate 0.1. All other parameters were kept as in the original paper. After each epoch, we update the dual variables (step 4), also using ADAM with a step size of 0.01. We take $c = 10^{-3}$. Both classifiers were trained over 300 epochs.

Without the constraint in (PVII), the resulting classifier is quite sensitive to gender: its prediction would changes for approximately 8% of the test samples if their gender were reversed (Figure 3). With the pointwise constraint, the classifier becomes insensitive to the protected variable in 99.9% of the test set, which is on the order of $1/\sqrt{N} \approx 0.008$. While the less strict ACE can also be imposed, it leads to slightly more sensitive classifiers (for $c = 5 \times 10^{-4}$, the classifier changes prediction in 0.2% of the test set).

Table 1: Preprocessing of the Adult dataset

| Variable names | Transformation |
| --- | --- |
| fnlwgt | Dropped |
| educational-num | Dropped |
| relationship | Dropped |
| capital-gain | Dropped |
| capital-loss | Dropped |
| education | Grouped the levels Preschool, 1st-4th, 5th-6th, 7th-8th, 9th, 10th, 11th, 12th |
| race | Grouped the levels Other and Amer-Indian-Eskimo |
| marital-status | Grouped the levels Married-civ-spouse, Married-AF-spouse, Married-spouse-absent |
| marital-status | Grouped the levels Divorced, Separated |
| race | Grouped the levels Other and Amer-Indian-Eskimo |
| native-country | Grouped the levels Columbia, Cuba, Guatemala, Haiti, Ecuador, El-Salvador, Dominican-Republic, Honduras, Jamaica, Nicaragua, Peru, Trinadad&Tobago |
| native-country | Grouped the levels England, France, Germany, Greece, Holand-Netherlands, Hungary, Italy, Ireland, Portugal, Scotland, Poland, Yugoslavia |
| native-country | Grouped the levels Cambodia, Laos, Philippines, Thailand, Vietnam |
| native-country | Grouped the levels China, Hong, Taiwan |
| native-country | Grouped the levels United-States, Outlying-US(Guam-USVI-etc), Puerto-Rico |
| age | Binned by quantiles (6 bins) |
| hours-per-week | Binned levels into less than 40 and more than 40 |

Figure 3: Classifier sensitivity on the Adult test set.

As we mention in the main text, due to the bound on the duality gap, the dual variables of (PVII) obtained in Algorithm 1 have a sensitivity interpretation: the larger their value, the harder the constraint is to satisfy [18]. Almost $96\%$ of the dual variables are zero after convergence, meaning that the constraint was tight for only $4\%$ of the individuals. In Figure 4a, we show the distribution of $\lambda > 0$ over the Adult training set. If we analyze the group with the largest dual variables (the $80\%$ percentile to be exact), we find a significantly higher prevalence of married individuals, non-white, non-US natives, and with a Masters degree (Figure 4b). Clearly, while attempting to control for gender invariance, the constrained learner also had to overcome other prejudices correlated to sexism in the dataset.

This situation even clearer in the COMPAS dataset. Here, the goal is to predict recidivism based on an individual's past offense data (see Table D.1 for details on the data processing). We use the same neural network as before trained over $400$ iterations using a similar procedure, but with batch size 256, primal learning rate $0.1$, and dual variables learning rate 2 (halved every 50 iterations).

(a)                                                    (b)

Figure 4: Dual variable analysis for Adult dataset: (a) distribution of the dual variables values and (b) prevalence of different groups among the $20\%$ training set examples with largest dual variables.

Table 2: Preprocessing of the COMPAS dataset

| Variable names | Transformation |
| --- | --- |
| age_cat | Dropped |
| is_recid | Dropped |
| is_violent_recid | Dropped |
| score_text | Dropped |
| v_score_text | Dropped |
| decile_score | Dropped |
| v_decile_score | Dropped |
| race | Grouped the levels Other, Asian, Native American |
| age | Binned by quantiles (5 bins) |
| priors_count | Binned levels into 0, 1, 2, 3, 4, and more than 4 |
| juv_misd_count | Binned levels into 0, 1, and more than 1 |
| juv_other_count | Binned levels into 0, 1, and more than 1 |

Unconstrained, it reaches an accuracy of almost $70\%$, but is sensitive to both gender, race, and gender $\times$ race (Table D.2). By including ACE constraints on these counterfactuals, we obtain a classifier that is now invariant to these variables.

Once again, the value of the dual variables capture insights into the different forms of biases existing in the dataset (Figure 5). If we do not include constraints on the cross-term counterfactuals, then the hardest constraint to satisfy is the gender-invariant one. Invariance to the Caucasian-Hispanic and Hispanic:Other counterfactuals is effectively "implied" by the other constraints, since their dual variables vanish. If we include all 13 counterfactuals, i.e., add the cross-terms between gender and race, then the cross-terms dominate the satisfaction difficulty, with the Male/Female $\times$ African-American/Caucasian dichotomy dominating over all others. What is interesting, however, is that the dual variable for the African-American/Caucasian counterfactual does not vanish, indicating the existence of a gender-independent race bias in the dataset. This does not occur with other combinations of the race factor. This type of combinatorial (gerrymandering) fairness is a serious challenge in fair classification [27].

## D.2 Robust learning

Although adversarial training has been successfully used to train robust ML models, it often leads to solutions with poor nominal performance, i.e., poor performance on original, clean data [30, 31, 44, 52, 53, 55]. To overcome this issue, [32] poses a constrained learning that explicitly trades-off nominal performance and performance against a worst-case perturbation. They propose an algorithm that optimizes over an upper bound of this robust constraint, leading to solutions that are simultaneously accurate on clean data and robust against input perturbations. Here, we follow a similar lead, but pose the problem as in (PIII) for a given adversarial distribution $\mathfrak{A}$ instead of optimizing of the worst

Table 3: Classifier insensitivity on the COMPAS dataset

| Counterfactual | Unc. (Acc: 69.4%) | ACE (Acc: 67.9%) |
|---|---|---|
| Male ↔ Female | 21.4% | 0% |
| African-American ↔ Caucasian | 10.86% | 0% |
| African-American ↔ Hispanic | 14.32% | 0.02% |
| African-American ↔ Other | 11.38% | 0% |
| Caucasian ↔ Hispanic | 9.11% | 0% |
| Caucasian ↔ Other | 6.54% | 0% |
| Hispanic ↔ Other | 3.08% | 0% |
| Male ↔ Female + African-American ↔ Caucasian | 28.84% | 0.02% |
| Male ↔ Female + African-American ↔ Hispanic | 27.47% | 0% |
| Male ↔ Female + African-American ↔ Other | 29.17% | 0% |
| Male ↔ Female + Caucasian ↔ Hispanic | 22.71% | 0% |
| Male ↔ Female + Caucasian ↔ Other | 24.27% | 0% |
| Male ↔ Female + Hispanic ↔ Other | 21.15% | 0% |

Figure 5: Dual variables of different counterfactual constraints for the COMPAS dataset.

possible one. This distribution can then be tailored to provide a smooth performance degradation instead of a worst-case robustness one.

To be concrete, consider the problem of training a ResNet18 [72] to classify images from the FMNIST dataset [73]. We reserve 100 images from each class sampled at random for validation. When trained without constraints over 100 epochs using the ADAM optimizer with the settings in [70] and batches of 128 images, it reaches it best accuracy over the validation set after 67 epochs. The nominal accuracy of this solution (over the test set) is $93.5\%$. However, when the input is attacked using PGD [30], it fails to classify any of the test images for perturbation magnitudes as low as $\varepsilon = 0.04$ (Figure 6a). In what follows, $\varepsilon$ indicates the maximum pixel modification allowed ($\ell_\infty$-norm of the perturbation) and we run the PGD attack using a step size of $\varepsilon/30$ for 50 iterations and display the worst result over 10 restarts, unless stated otherwise.

A first attempt is then to use PGD with $\varepsilon = 0.04$ to sample from a hypothetical adversarial distribution and constrain its performance against that distribution as in (PIII). Though the adversarial distribution is now dependent on the model $\phi$, by using a smaller learning rate for the dual variables, $\phi$ can be considered almost static for the dual update and we have observed no instability issues in practice. To accelerate training, we use a much weaker attack running PGD without restarts for only 5 steps with step size $\varepsilon/3$. Notice from Figure 6a that when training against $\varepsilon = 0.04$ ($c = 0.4$), the resulting classifier trades-off nominal performance (now $88\%$) for adversarial performance (now $85\%$). However, as the strength of the attack increases, the performance of the classifier deteriorates abruptly: for $\varepsilon = 0.08$, it is down to $9\%$. Increasing the training adversarial strength to $\varepsilon = 0.1$ ($c = 0.7$) yields

(a)

(b)

Figure 6: Robust constrained learning (FMNIST): (a) Accuracy of classifiers under the PGD attack for different perturbation magnitudes and (b) distribution of $\varepsilon$ used during training.

(a)

(b)

Figure 7: Robust constrained learning (CIFAR-10): (a) Accuracy of classifiers under the PGD attack for different perturbation magnitudes and (b) distribution of $\varepsilon$ used during training.

a more robust classifier, albeit at the cost of a lower nominal accuracy ($84.6\%$). Still, the performance degradation remains quite abrupt.

This issue can be fixed by training against using a hierarchical adversarial distribution. Explicitly, we build the adversarial distribution $\mathfrak{A}$ as

$$\Pr\left(\mathfrak{A}\right) = \Pr\left(\mathfrak{A} \mid \varepsilon\right)\Pr\left(\varepsilon\right), \tag{51}$$

where $\Pr\left(\mathfrak{A} \mid \varepsilon\right)$ is induced by an adversarial attack of magnitude at most $\varepsilon$ (in our case, PGD) and $\Pr\left(\varepsilon\right)$ denotes a prior distribution on the magnitude of the attacks. In Figure 6a we take $\varepsilon \sim 0.25 \times \text{Beta}(3, 8)$ (Figure 6b). Notice that even though the mean value of the perturbation is approximately $0.07$, the resulting classifier has a nominal performance close to $87\%$ and retains a $67\%$ accuracy for perturbations of magnitude up to $0.12$.

Similar results are obtained when training a ResNet18 [72] to classify images in the CIFAR-10 dataset. The training was performed as above, once again reserving 100 random images from each class sampled for validation. The unconstrained classifier trained over 100 epochs reached it best accuracy over the validation set after 82 epochs, which corresponds to a nominal test accuracy of $85.4\%$. However, when the input is attacked using PGD [30], the accuracy falls to $5\%$ already for $\varepsilon = 0.01$ (Figure 7a). When using the fixed $\varepsilon$ training method described above, we once again observe a trade-off between nominal accuracy and robustness. This can, however, be improved using the hierarchical training technique from (51). Taking $\varepsilon \sim 0.1 \times \text{Beta}(3, 10)$, such that $\mathbb{E}\left[\varepsilon\right] = 0.02$, we obtain the same nominal accuracy as for the fixed-$\varepsilon$, but improve the robustness for higher perturbation values.