[Reviews · NeurIPS 2020]

Review 1

Summary and Contributions: The paper investigated the constrained learning based on the probably approximately correct (PAC) learning framework. Authors proposed the framework of constrained learning and generalizsed theory and, demonstrated how to approximate such learning in practice.

Strengths: Constraint learning was nicely framed and it was easy to follow the problem and ideas. Based on the empirical Lagrangian, the constrained learning solution was proposed and authors showed that this is compatible.

Weaknesses: The compatibility of the new constraint learning algorithm was shown through simulation studies. For the first simulation study, the constraint is on the KL for male and female is under some threshold, c. Why would you want to fit a parametric model with small difference (or zero difference) between gender? For data analysis purpose, wouldn’t you want to have a model in which characterize gender difference? This framework is useful when there are clear constraints due to the nature of problem. When there is no clear constraints, can we use this algorithm to improve a better fit or investigate particular feature of data?

Correctness: Although this is not my area, I didn’t find any major fault.

Clarity: It was easy to follow the paper and the structure was clear.

Relation to Prior Work: Literature review was provided.

Reproducibility: Yes

Additional Feedback:


Review 2

Summary and Contributions: The paper elaborates a theory for PAC learning under constraints. Authors show that PAC and PACC learning are equivalently hard, and derive a generalization bound for PACC. They also develop a method for learning and show experiments that support their theory.

Strengths: 1. Claims and experimental set up are technically correct, though the latter is detailed in the appendix. 2. The line of work is novel since there is scarce work in PAC learnability under constraints. The significance of the theory might be somewhat limited as it shares the same limitations of unconstrained PAC, where there is more advances like PAC-Bayes and its many variants. 3. The work is of high relevance to the learning community as it serves as first steps toward formalizing learnability under constraints.

Weaknesses: In my opinion the main weakness/limitation is the dependance of the generalization bound on the optimal lagrange multipliers (dual variables). That is, it unclear to me how could we compute the number of samples without knowing them. The appeal of many generalization bounds is that one could compute the sufficient number of samples to attain certain desired error. I would appreciate if authors can comment on this concern.

Correctness: Yes, all claims and experimental set up are sound.

Clarity: Yes, the paper delivers its message clearly.

Relation to Prior Work: Authors do a good job relating and contrasting to prior work.

Reproducibility: Yes

Additional Feedback: Some minor comments: * What does the acronym CSL stand for? * The norms in result of Theorem 2 (equation 6), should both be || ||_1 or does L_1 denotes another norm? * There is a bracket missing in (3a) === UPDATE: After reading author's response, I am leaving my score unchanged. I consider the work to be a good contribution to the theory of constrained learning.


Review 3

Summary and Contributions: The paper extends the probably approximately correct (PAC) learning framework to include constraints on the learned hypothesis. This new framework, coined PACC is shown to be as hard as a corresponding PAC problem (without constraints). A practical algorithm for PACC learning is discussed.

Strengths: Putting forward an extension of the PAC set up by taking constraints explicitly into account seems an interesting idea. The paper also illustrates how to use this extended framework for implementing fair and robust ML methods.

Weaknesses: - I suggest adding a detailed outline of the paper in the introduction section - the contributions relative to existing work should be spelled out more explicitly in Section 2 - the quality of presentation and clarity should be improved - it is not made clear how the theory developed in this paper allows interpreting the numerical results,e .g. "Doing so yields nominal accuracies of 87% with adversarial accuracies of 82% at 0.04, 75% at 0.08...."

Correctness: Probably.

Clarity: the wording/clarity might be improved, e.g., - "...we can soften the worst-case requirements..." maybe use weakened instead of soften - i guess in (3a) and (3b) are some typos. e.g., what is the index i and j there ? - "PACC Learning is as Hard..." would it make sense to use "is not harder than PAC" instead of "..is as hard ..." - "...that PAC and PACC learning are equivalent problems." in which precise sense are they equivalent? - "Although we know this dual problem is not related to (PIV)..." how is it related precisely ? - how is epsilon related to epsilon_0 in Theorem 2? - what is the input and what is the output of Algorithm 1 - maybe you could state Algorithm 1 earlier in the paper to improve the readability - "... if perturbations as small as 0.04 are allowed" - "...the dual variables have a sensitivity interpretation: the larger their value, the harder the constraint is to satisfy [18]" can we make this interpretation precise by some bound ?

Relation to Prior Work: the offerings made by this paper relative to existing work could be made more explicit in Section 2.

Reproducibility: No

Additional Feedback: i could not find the details for how step 3 of Algorithm 1 is implemented in the experiments.

[Author Response · NeurIPS 2020]

We thank the reviewers for their time and for consideration. They will find responses to their specific points below.

**Reviewer 2.** While we use fairness and robustness to illustrate the use of constraints in learning, the main results of the paper (Thm. 1–3) are independent of these applications or the experimental settings. Still, we believe they are relevant. Models invariant to gender, e.g., can be used for fair decision-making and, in a more data analysis context, to seek patterns masked by gender or identify, through the dual variables, the features (phenotypes) more sensitive to gender. We illustrate such analyses in Section D of the appendices. As the reviewer notes, this framework has applications beyond model structure and preliminary results suggest that pointwise, per data point constraints can be used to improve fit. However, this is beyond the scope of this work that focuses on understanding what can be achieved when learning under constraints. Due to space limitations we could not address these points in the main body of the paper, but we intend to use the extra page in the final manuscript to bring experimental details from the appendices. If this does not fully address the reviewer concerns about the broader impact of the manuscript, we can work on expanding that section for the camera-ready.

**Reviewer 3.** We agree that, being a new area of research, constrained learning theory still has many open questions, such as understanding the effect of constraints on other learning models (e.g., structured complexity and PAC-Bayes) and learning forms (e.g., reinforcement learning). We will include these discussions in the camera-ready. Still, this work already shows that constrained learning is fundamentally different than PAC learning, especially dual constrained learning where the dependence on $\boldsymbol{\mu}$ and $\boldsymbol{\lambda}$ appears. Nevertheless, they affect only the *parametrization error* $\epsilon_0$, which is independent of the sample size (Def. 3). Hence, $\epsilon_0$ is an intrinsic limitation of the parametrized learning problem and sample sizes can always be estimated relative to this lower error bound. Alternatively, there is a well-known upper bound from optimization theory, which for the case of $[0, B]$-bounded objective yields $\|\boldsymbol{\mu}^\star\|_1 \leq Bs^{-1}$, where $s = \min_i \mathbb{E}\left[\ell_0\left(f_{\boldsymbol{\theta}'}(\boldsymbol{x}), y\right)\right] - c_i$ is the smallest slack for any strictly feasible solution $\boldsymbol{\theta}'$ (i.e., such that $s > 0$). If the output $\boldsymbol{\theta}^{(T)}$ of Alg. 1 is sufficiently in the interior of the feasible set, bounds for the proof of Thm. 2 yield $s \leq \min_i \left[c_i - M\nu - B\zeta(N_i) - \frac{1}{N_i}\sum_{n_i=1}^{N_i} \ell_i\left(f_{\boldsymbol{\theta}^{(T)}}(\boldsymbol{x}_{n_i}), y_{n_i}\right)\right]_+$. While more tractable, we do not think that this result is convenient as it requires the constraints to be quite loose for the bound not to vanish. We therefore did not include these details in the manuscript, but will discuss this limitation more clearly in the camera-ready. Also, we use CSL to mean "Constrained Statistical Learning" and write $\|\cdot\|_1$ to denote the $\ell_1$-norm (i.e., on a space of sequences) and $\|\cdot\|_{L_1}$ to denote the $L_1$-norm (i.e., on a space of functions). This distinction occurs because the dual variable of the $j$-th pointwise constraint is actually a continuous (as opposed to discrete) function.

**Reviewer 4.** The reviewer raises good points to improve the clarity of the paper. We intend to use the extra page of the camera-ready to address them as well as bring part of the numerical experiments discussion from the appendices. In particular, we will emphasize how the contributions of the paper provide generalization guarantees and therefore ensure that an accurate, robust classifier trained using our constrained method will also perform well in testing (as evidenced by Figures 4 and 5). Addressing the reviewer's questions more specifically, the indices in (3) refer to the constraints in (P-CSL), i.e., (3a) relates to the $i$-th average constraint of the learning problem and (3b) relates to the $j$-th pointwise constraint. The reason we write "PACC Learning is as Hard as..." is because Thm. 1 is both necessary and sufficient. That PAC is not harder than PACC is trivial since PACC includes PAC in (2). But Thm. 1 also proves the converse. We then say these forms of learning are equivalent in the sense that a hypothesis class is PACC learnable if and only if it is PAC learnable (Thm. 1). As for the relation between (PIV) and its dual (D̂-CSL), recall that (PIV) is non-convex so that its dual only provides a lower bound on its value (weak duality). However, Thm. 2 shows that (D̂-CSL) provides both lower and upper bounds for the original (P-CSL). In Def. 3, $\epsilon$ and $\epsilon_0$ are not related: $\epsilon$ is the *estimation error* while $\epsilon_0$ is the *parametrization error*. Indeed, $\epsilon_0$ is independent of the sample size and depends only on the learning problem ($M$ and dual variables) and the richness of the parametrization ($\nu$). Although the parametrization error is treated separately in unconstrained learning, it is not possible to untangle it for (dual) PACC learning (see proof of Thm. 2). In terms of Alg. 1, its input is a sample set and its output is a parameter vector $\boldsymbol{\theta}^{(T)}$ that describes the classifier. Dual variables can also be extracted for analysis (see Section D). In these analyses, we leverage the sensitivity interpretation of the dual variables, which is formalized by the fact that the optimal dual variables are *subgradients* of the empirical Lagrangian at the optimal primal solution. In fact, we show that this is (approximately) true even at the approximate oracle from Assumption 4 (see Lemma 1 in Section C). To be more specific, it means that the larger the dual variable, the more the optimal value would increase if we were to tighten that constraint, indicating that the constraint is "harder" to satisfy. We use a mini-batch SGD algorithm for step 3 of Alg. 1 (details can be found in the appendices). Note that the results of Thm. 3 do not depend on this method and there may be better suited approaches in certain problems, but this was a natural choice for the logistic and CNN classifiers used in the experiments. While we will certainly expand Section 2, we note that, to the best of our knowledge, this is the first paper to explicitly analyze constraints in the context of (PAC) learning theory. Previous work focus on specific constraints, e.g., rates, and provide guarantees for specific algorithms that typically yield randomized solutions. In contrast, this paper provides algorithm-independent results (Thm. 1 and 2) for deterministic learners. It also considers pointwise constraints that are important in domains such as fairness.

[Meta-Review · NeurIPS 2020]

The authors propose an extension to PAC learning by considering explicitly constraints that are imposed in particular settings. These could be, for example, fairness constraints or robustness constraints. The theory and the experimental set up are technically correct. The work is of high relevance to the machine learning community as it incorporates into a nice formalism the constraints that are often imposed on the learning algorithms.